# Mesoscale neuronal granular trial variability in vivo illustrated by nonlinear recurrent network in silico

Guihua Xiao [1,2,3,5], Yeyi Cai[2,3,5], Yuanlong Zhang[2,3], Jingyu Xie[2,3], Lifan Wu[2,3], Hao Xie[2,3], Jiamin Wu [1,2,3,4] ✉ & Qionghai Dai [1,2,3,4] ✉

Large-scale neural recording with single-neuron resolution has revealed the functional complexity of the neural systems. However, even under well-designed task conditions, the cortex-wide network exhibits highly dynamic trial variability, posing challenges to the conventional trial-averaged analysis. To study mesoscale trial variability, we conducted a comparative study between fluorescence imaging of layer-2/3 neurons in vivo and network simulation in silico. We imaged up to 40,000 cortical neurons' triggered responses by deep brain stimulus (DBS). And we build an in silico network to reproduce the biological phenomena we observed in vivo. We proved the existence of ineluctable trial variability and found it influenced by input amplitude and range. Moreover, we demonstrated that a spatially hetero-geneous coding community accounts for more reliable inter-trial coding despite single-unit trial variability. A deeper understanding of trial variability from the perspective of a dynamical system may lead to uncovering intellectual abilities such as parallel coding and creativity.

Repeated stimulus trials have long been a cornerstone methodology in neuroscience for delineating neural mechanisms, yet they are often plagued by significant trial-to-trial variability. Unlike digital computers, the bio-neural network generates diverse responses while receiving identical sensory input, particularly in large-scale neuron recordings. Understanding the origin and dynamical patterns of trial variability has been a longstanding puzzle for the neuroscience community.

With the advancement in recording technologies, large-scale neural imaging[1,2] has brought an unprecedented number of simulta-neously recorded neurons, expanding up to the whole cortex of the rodent. In contrast to electrophysiological recording[3], mesoscale fluorescence imaging, can cover a larger horizontal field of view (FOV) while still maintaining single-neuron resolution[4–7]. To some extent, this approach has saved the effort of pre-location of targeted neuron populations such as tentative electrode implantation or intrinsic imaging[8]. However, with the rapid growth of co-captured brain regions, spontaneous neural activities quickly overwhelmed task-relevant neural signals, constituting the major explained variance in cortical activity[9–11]. Moreover, it has been reported that task variables are sparsely coded by neurons distributed across distant brain areas[12,13], exhibiting varied information density. The increase in signal sources comes with the side-effect of a decreased task-relevant signal-to-noise ratio and reduced inter-trial consistency, attributed to signal aliasing. Traditionally, this variability was viewed as noise obscuring the task-relevant signals, to be minimized by averaging neural responses across trials. This approach also implies that significant valid information could be averaged out, potentially resulting in incorrect conclusions. This raises a pivotal question for the neuroscience com-munity: Can trial variability be controlled or fully eradicated? If not, how does the brain manage to achieve stable coding amidst such variability?

[1]Beijing National Research Center for Information Science and Technology, Tsinghua University, Beijing, China. [2]Department of Automation, Tsinghua University, Beijing, China. [3]Institute for Brain and Cognitive Sciences, Tsinghua University, Beijing, China. [4]IDG/McGovern Institute for Brain Research, Tsinghua University, Beijing, China. [5]These authors contributed equally: Guihua Xiao, Yeyi Cai. ✉e-mail: wujiamin@tsinghua.edu.cn; qhdai@tsinghua.edu.cn

Meanwhile, the progress in dynamic systems fosters our understanding of the self-organization property of the brain[14–18], which is often considered the origin of intelligence[19] and creativity[20]. From the perspective of neural dynamics, the brain state travels along the energy landscape specified by the dynamical property of cells and their interactive networks. External input to the system serves as a controlling command, which regulates the brain towards a specific local attractor[21–24]. The temporary equilibria underlie animal behavior such as decision-making[25] and state change[26,27]. However, the exact neural space trajectory varies from time to time depending on the initial state of the network[28], as well as the stochastics of the near-critical property of the neural network[29]. When extending the observation scope to cortex-wide neurons, the conventional sensory input becomes less influential in neural populations. Low-resolution imaging modalities, such as wide-field imaging, largely mix local area intensities, obscuring individual neuron properties. The dynamic study in single brain areas based on designed task variables became less efficient. Several requirements must be met before analyzing cortex-wide dynamics. First, there should be parameterized and broadcast stimulation that could mobilize a wide range of neurons. Secondly, neurons must be captured simultaneously in a single trial, without trial average and cross-trial stitching. This has raised challenges to conventional imaging modalities and experiment designs. Misunderstandings of task-irrelevant signals have intimidated neuroscientists from employing large-scale neural recording techniques. Great amounts of information are neglected through the strict brain-area screening procedure for highly mission-selective neurons or performing trial-averaging[30]. Therefore, understanding cortex-wide dynamics under trial variability remains an unsolved problem, due to the lack of suitable imaging techniques and synchronous parameterized paradigm.

To answer these questions, we designed a paradigm that delivers sequentially repeated input to the deep brain nucleus and records large-scale cortex-wide single-neuron responses. The deep brain electrical stimulation (DBS) is synergized with mesoscale optical imaging. Up to 40,000 layer-2/3 neurons in each mouse are captured in a single trial. Among them, thousands of neurons are effectively tuned, making it possible to find single-trial variability properties in the mesoscale range. To dissect and understand the origin of this variability, a rate-based dynamic network model was built in silico, receiving arbitrary input which was tuned in amplitude and projection range. By tuning and comparing the experiment in vivo and the model in silico, we found similar trial variability properties between these counterparts. The key factors influencing trial variability were quantitatively identified. We demonstrated that the inherent trial diversity is an intrinsic feature of the nonlinear recurrent network even under globally intensive stimulation and a low noise assumption. With the help of in silico simulation, we also depicted the energy landscape of the network under stimulation, which explains the dynamic behavior in the neural space. To understand how imperfectly tuned neurons constitute a functional coding community, we recorded cortical network responses to repeatedly presented visual inputs. The regions surrounding the primary visual cortex in mice have been linked to various types of visual stimuli, including drifting patterns[31] and natural images[32]. Neuronal ensembles in these areas have been shown to encode visual stimuli effectively[33] and maintain stability over several weeks[34]. Recent studies employing two-photon calcium imaging have demonstrated that the activities of neuronal populations provide a more accurate prediction of visual stimuli. This suggests a robust and durable encoding capacity within these cortical areas[31]. The trial variability of single coding units exhibited a spatially heterogeneous pattern. While highly reliable neurons are concentrated in the visual cortex, there are also dispersed coding neurons in the periphery. At a certain point, including more neurons in the imaging field induces higher irrelevant information and leads to decreased decoding performance. This observation is verified by a simple topological assumption of input weight in silico. Identifying functional neurons among the whole cortex or performing dimension reduction are effective ways of decoding the concerned information with significantly fewer dimensions. The comparative study between the in vivo experiment and the in silico simulation offers a comprehensive way of studying trial variability, suggesting that apart from controlling the experiment environment and reducing external noises, further challenges lay in separating functional neural populations and digging undesigned variables. By dissecting the complexity of trial variability, we can uncover the coding strategies concealed within these seemingly unrelated signals, thereby deepening our understanding of bio-intelligence and the manifestation of consciousness in the brain. By intelligently regulating this coding strategy, we can also provide great value for the promotion of intelligent brain–computer interface technology.

## Results

### A synergistic system of mesoscale fluorescence imaging and DBS

To effectively tune neurons expanding multiple functional cortical areas, we started with DBS[35] as the stimulus modality under specific parameters (100 μs pulse duration, 100 Hz frequency, 1 s stimulation time, and 300 μA amplitude). DBS is a widely accepted technique used to modulate abnormal neural activity by causing depolarization or hyperpolarization of neurons near the electrode[36]. The neuron firing rates would be modified either by enhancing or inhibiting their activity. Additionally, DBS can stimulate the release of various neurotransmitters, such as dopamine[37,38]. Clinically, DBS is extensively employed to treat Parkinson's disease by modulating the abnormal firing patterns in the subthalamic nucleus[39], to reduce symptoms of movement disorders, influencing mouse behavior. DBS can modify the functional connectivity between different brain regions, resulting in either increased synchronization or desynchronization across the network. Chronic DBS has also been shown to induce neural plasticity[40], potentially altering the brain's response to experimental conditions over time. In our study, mice received DBS on a fresh day to prevent trial variability caused by changes in network connectivity or neural plasticity. Behavior was also well defined to assess its impact on trial variability on our results.

To simultaneously deliver electrical stimulus and capture neural responses, electrical stimulation regulation, and fluorescence imaging are jointly used as stimulation and observation tools (Fig. 1a). Platinum-iridium wires electrodes are designed[41,42] and implanted in sub-cortical Cla nucleus to both record deep brain signals and deliver parameterized stimulus. The electrodes are exquisitely arranged to avoid occlusion in the optical path for fluorescence imaging (Supplementary Fig. 1a, b). For calcium imaging at single neuron resolution, we utilized our previously developed mesoscale imaging system RUSH[43] which covers a $10 \times 12\,mm^2$ field of view, with a uniform resolution of about 1.2 μm.

Transgenic mice expressing GCaMP6f in layer 2/3 (Rasgrf2-2A-dCre/Ai148D)[44] are used to visualize neural activities, which shows a much higher signal-to-noise ratio than Thy1-GCaMP6f transgenic mice due to the reduction of background fluorescence (Supplementary Fig. 1d, e). The cranium is carefully removed and replaced with a piece of curved cover glass to expose the cortex surface[45]. During the experiment, the mouse is head-fixed with customized holders and free to move on a treadmill. Multi-modality behaviors, including running speed, pupil size, and facial expression, are recorded by sensors and cameras (Supplementary Fig. 2a and Supplementary Movie 1). Calcium activities at the single neuron level are extracted and filtered[46,47] from mesoscale imaging camera arrays. The locomotion state and pupil diameter exhibit good synchronicity with the calcium dynamics (Fig. 1b).

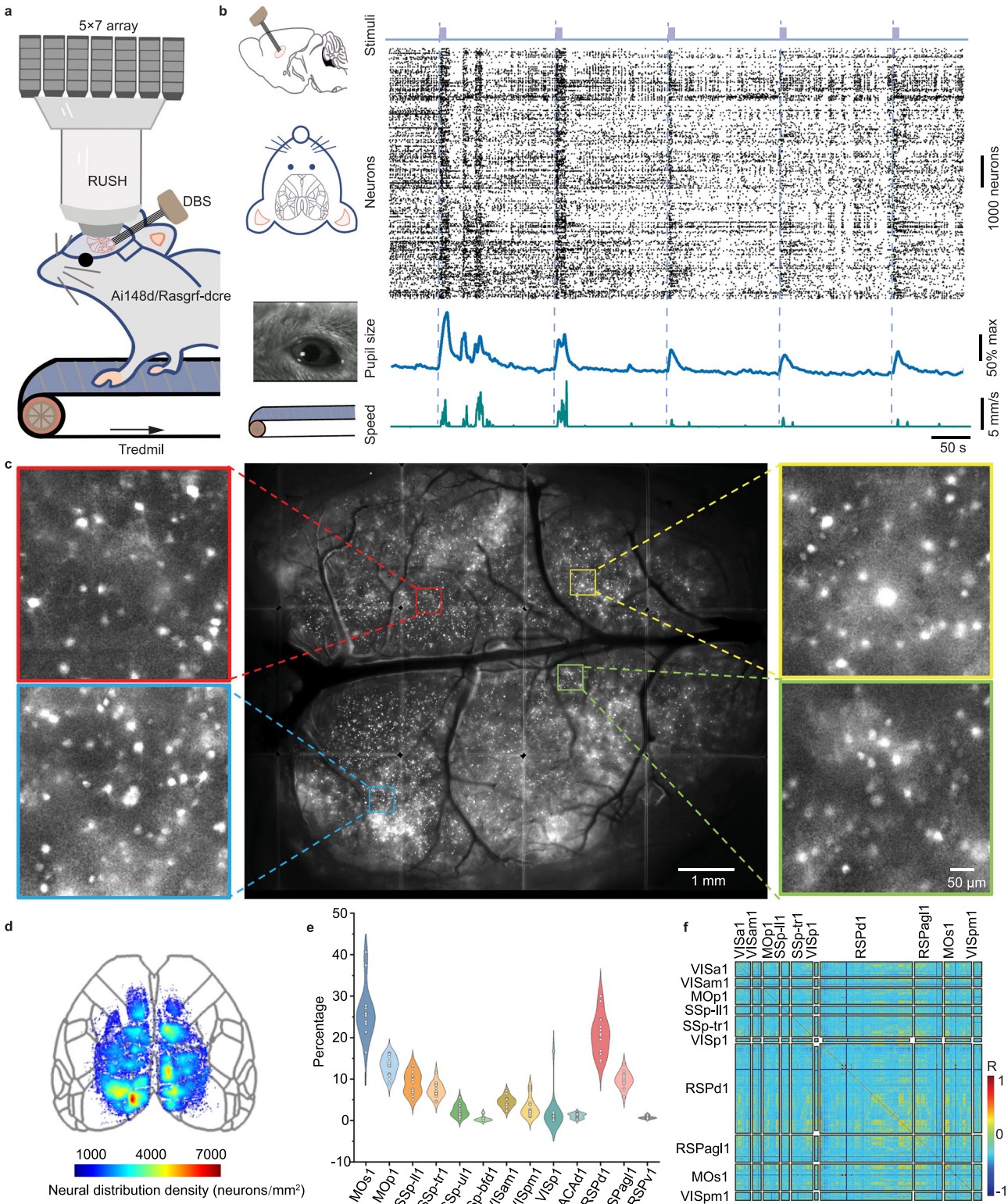

**Fig. 1 | Synergism of cortex-wide single-cell resolution calcium imaging and electrical stimulation. a** Schematic drawing of the experiment setup by combining RUSH imaging system and deep brain stimulus (DBS) recording. **b** Multimodal signals detected in the head-fixed mice include the synchronous changes of pupil area, running speed, and calcium traces of neurons. The stimulus electrode would deliver a biphasic waveform to Cla several times. **c** Example of cortex-wide calcium imaging at single-cell level obtained by RUSH system. Enlarged views of the cortical areas marked by the boxes show clear cellular profiles of individual neurons. This neuron distribution result is similar in other mice (*n* = 15 mice). **d** Approximately 40,000 active neurons across wide cortical regions were detected in a typical mouse. **e** Distributions of neurons in 12 cortical regions (dots are the percentage of neuron number in the corresponding area from individual mice, *n* = 15 mice). **f** Correlation analysis of the cortex-wide activities at the single-cell level. Source data are provided as a Source Data file.

Up to 40,000 active neurons can be captured in each single trial, with a sampling rate of 10 Hz. The uniform resolution ensures a high-quality identification of single neurons (Fig. 1c and Supplementary Movie 2). Since the data throughput is up to 2TB per 10 min, a neuron extraction pipeline has been developed to detect neurons with background rejection, stitch patched views, and reject false positive traces induced by vascular dynamics[43] (Supplementary Fig. 2b). The extracted single neuron footprints are registered to Allen CCF using a retinotopic map of visual cortex and cranial anatomical landmarks acquired during surgery (Supplementary Fig. 1c). The imaged brain areas span more than a dozen CCF areas, including visual, somatosensory, motor, posterior parietal, anterior cerebral, and retrosplenial cortex (Fig. 1d, e, Supplementary Fig. 2c, d and "Methods"). We conducted a correlation analysis of cortical brain activities. The resulting correlation matrix contains near-zero correlation coefficients, which proved the low-cross-talk property of the RUSH system compared with low-resolution wide-field imaging (Fig. 1f and Supplementary Fig. 3a). Mesoscale cortex imaging, combined with a detailed dissection of neighboring neurons' activities, gave us a more precise description of single-trial neural activities and their cross-trial variability.

A prominent aspect of dissecting single neurons from populational activity is the decreased spatiotemporal correlation. In addition to the temporal trial variability for each neuron, the spatial distribution of single trial response also exhibited great variability. Low-resolution wide-filed imaging is severely disrupted by background fluorescence fluctuation due to the lack of hardware and software unmixing. The correlation distribution among different neurons within a single trial, as well as the cross-trial correlation of single neuronal signals obtained by RUSH (Supplementary Fig. 3b, c) are significantly lower compared with low-resolution imaging modalities. With the help of mesoscale single-neuron resolution imaging, we achieved the faithful representation of divergent tuning curves of neurons, which gives rise to a neural space with a higher dimensionality and supports the coding of the complex physical world.

### Ineluctable single-neuron and population-level trial variability under parameterized nucleus stimulation

It has been reported that projection from Cla to the cortex area is complex and distributed in a wide range[48,49]. Compared with functional sensory input, DBS serves as passive stimulation and travels through relatively short neural circuits before reaching the cortex[42]. DBS therefore avoids some task engagement variables influencing the trial variability[50], such as attention[51], engagement[52], arousal[53], or eye movements[54]. Moreover, unlike natural images or sounds, DBS could be stably parametrized and controlled via stimulation amplitude and frequency. Therefore, we chose DBS as a dispersed and strong stimulation to induce cortex-wide neural activities.

Under the considerations above, a repeated DBS paradigm is designed to study the trial variability of mesoscale cortex neurons. Micro-level biphasic electricity is delivered via the flexible electrode implanted in the Cla nucleus. The stimulus energy is carefully limited to prevent tonic-clonic seizures. The interval between pulses spans several minutes with a pseudo-random variation for the mice to recover fully to the resting state and reduce anticipation (Fig. 2a). Neurons exhibited a rapid and distinct response to the stimulation, with either activation or deactivation reactions (Fig. 2b and Supplementary Fig. 4a, b).

A direct observation of the cortical responses under DBS suggested an ineluctable trial-variability both at single-neuron resolution and at the low-resolution wide-field imaging scale (Fig. 2c). Although the consistency of the DBS input is accurately controlled, the resulting responses are diverse between trials. Although neurons in a specific area may exhibit a populational response tendency, the exact response of each neuron varies from time to time. The imperfection indicated

that in a complex nonlinear network, trial variability may be an intrinsic property that is essentially unavoidable. Also, we found that neighboring neuron pairs could exhibit completely contradictory tuning properties of the same stimulation (Supplementary Fig. 4b), suggesting the complexity of the neural granular network and the necessity of using single-neuron resolution imaging.

We then investigated the tuning effect of the DBS. We found that among all neurons, most neurons are slightly tuned, and still a minority of neurons receive a stronger impact from the electrical stimulation. The distribution of the $z$-scored neural fluorescence intensity changes is approximately centered at zero. The low-resolution wide-field result has a higher modulation effect after stimulus, which may be a result of background neuropil activities (Fig. 2d). DBS also modulates the functional connectivity between neurons (Fig. 2e, f). This may also be a result of collective firing or inhibition. Another significant effect of external stimulation is that it increases the inter-trial population vector correlation at single-neuron level (SN) and wide-field level (WF) (Fig. 2g). After the stimulation onset (SN post or WF post), the spontaneous uncorrelated neural activities are replaced with regulated activities specified by the external input pattern and the internal connectivity of the network, which corresponds to a local attractor in the neural space.

### Stimulus-shaped attractor model to explain neuronal granular trial variability

We next aimed to formulate a simple but effective forward model, integrating our previous findings of distributed projection from the nucleus to the cortex, sparse functional connectivity between neuron pairs, and the suppression of chaotic trial variability by external stimulation. We decided to follow the line of recurrent neural network model, which has been used in various studies to describe the dynamics of neural populations[22,26,55] (Fig. 3a and "Methods"). In our model, we have imposed minimal assumptions in our simple RNN model to show the trial variability as a naturally emerging property in the high-dimensional dynamic system. As previously described[25], neurons are randomly connected, following an approximately Gaussian distribution, which is also proved via the functional connectivity matrix, and the connectivity was fixed during stimulation, because the paradigm does not include learning or long-term synaptic development. Following the observation of sparse and distributed projection from the nucleus to the cortex, we set the external input to a certain range of input neurons. The values of the projection matrix are distributed mainly around zero, where a minority of neurons receive strong positive direct input, and most neurons are mildly projected or indirectly influenced. Although the external input is positively projected to the input neurons, the inhibited neuron community naturally emerges from the negative internal connectivity between neurons (Fig. 3b). Specifically, an activated first-layer neuron may transmit its positive activity to the next layer through a negative synaptic connection of an interneuron, leading to the deactivation of downstream neurons. In the following sections, we aim to find various similar statistical, behavioral, and coding properties between mesoscale neural populations in vivo and the single neural granular dynamic model in silico.

Like the in vivo experiment, we delivered a repeated external input to the network. At the single neuron level, we found neurons are imperfectly tuned (Fig. 3c and Supplementary Fig. 4b) both in silico and in vivo. The trial averaged results show an apparent activation or deactivation effect, but the extent of which varies between trials, even under extremely low noise assumption in the model (Supplementary Fig. 5a). The change of noise term in the model does not directly lead to the change of trial variability (Supplementary Fig. 5c, d). This indicates trial variability may be an intrinsic property of any self-organized network with high-dimensional nonlinear connections, which happens to be the basic property of the brain.

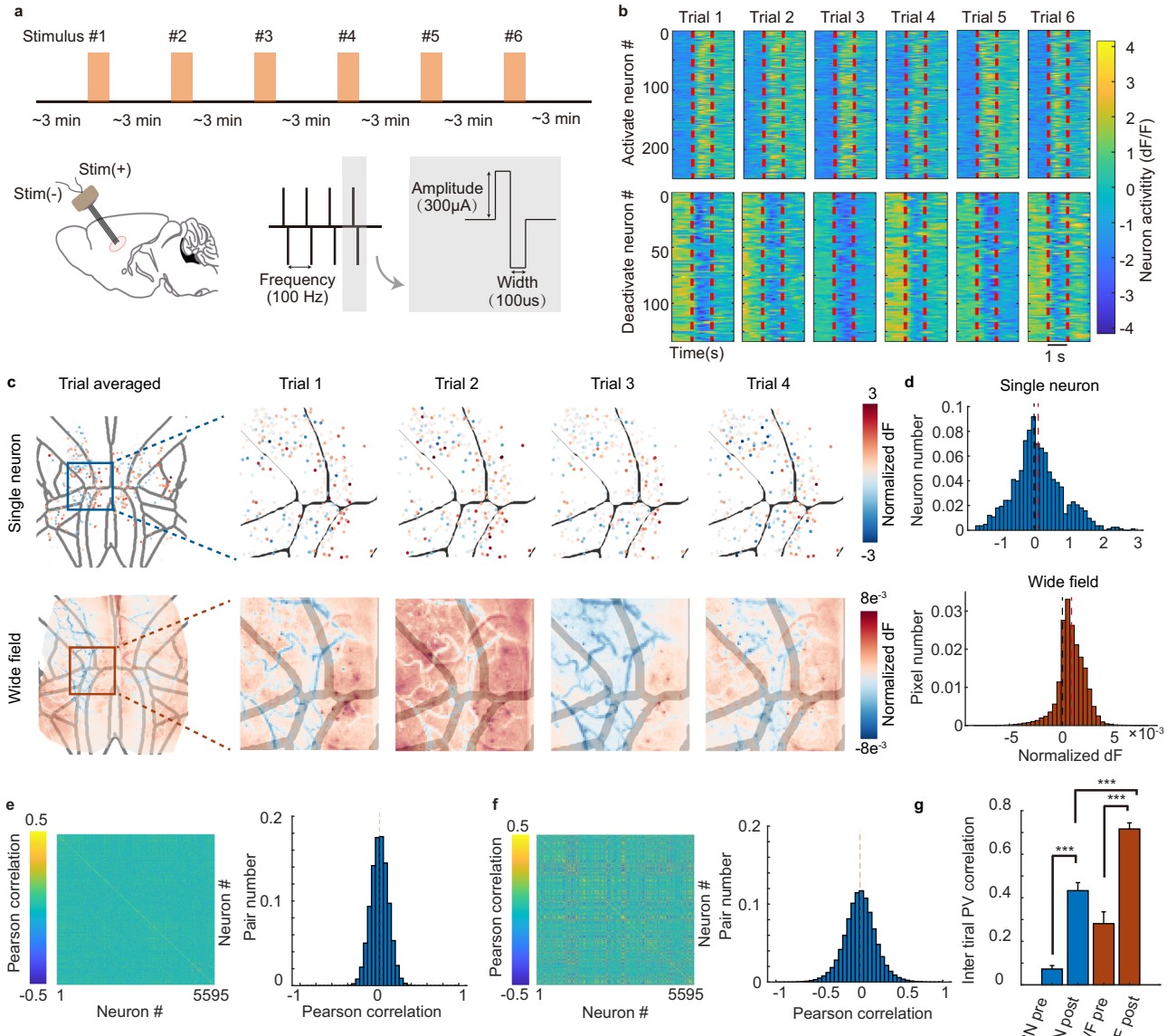

**Fig. 2 | Cortex-wide trial variance with deep brain stimulation. a** Diagram of deep brain stimulation with parameterized waveform. **b** An example of the neural response of electrical stimulus tuning. Different neural groups are oppositely tuned, some are activated (top) and others are deactivated (bottom). **c** Cortex-wide spatial response pattern of the same stimulation, on the scale of a single neuron (top) and wide-filed imaging (bottom). **d** The distribution of change of $z$-scored fluorescence intensity both in the single neuron modality (top) and the low-resolution wide-field modality (bottom). The black dashed line and the red dashed line show the value 0 and the mean response of neurons/pixels. **e** Trial-averaged

functional connectivity between neuron pairs evaluated by Pearson correlation in a 2-s time window before the DBS onset (left), and the distribution of the correlation (right). **f** Same as **e** but for a post-stimulation time window. **g** Comparison of population vector (PV) correlation pre- and post-stimulation at single-neuron level (SN) and wide-field level (WF). Three mice, $n = 45$ trial pairs. Paired two-sided Wilcoxon test. $p = 5 \times 10^{-9}$ between SN pre and SN post, $p = 1 \times 10^{-7}$ between WF pre and WF post, $p = 6 \times 10^{-9}$ between SN post and WF post. $^{***}p < 0.0005$; error bars: SEM. Source data are provided as a Source Data file.

The neural spatial intensity map of neural responses shows the propagation of electrical signals across the wide cortex (Supplementary Fig. 6a). We also found that the external input does not necessarily tune the same group of neurons in each trial. The number of repeatedly responsive neurons decreases with the number of trials concerned (Fig. 3d and Supplementary Fig. 4c, d) both in empirical observation and in simulated networks.

Previous studies[28] suggested that the initial state of animal behavior and the corresponding neural activities are affecting factors of cross-trial repeatability. To demonstrate this assumption, we divided the initial state of the mice based on the inspection of behaviors captured via an infrared camera. According to the total moving energy value of the face and limbs of the mice before the onset of stimulation,

the initial states were divided into two classes, locomotive and quiet (Fig. 3c and Supplementary Fig. 6b). In the recurrent model, considering that behaviors are often encoded by a low dimensional manifold in the neural space[56], we simplify the situation by considering the second principal component of the neural network as our interested behavior index outcome of the mice. Trials were then segregated into two types using an unsupervised clustering method. The inner- and inter-class correlation between trials was calculated both in empirical data and in simulated networks. The result indicated a higher correlation index between the same class trials compared to inter-class (Fig. 3e and Supplementary Fig. 5b). The analysis supports the idea that the initial state of the neural population could efficiently influence the outcome of the same stimulation input.

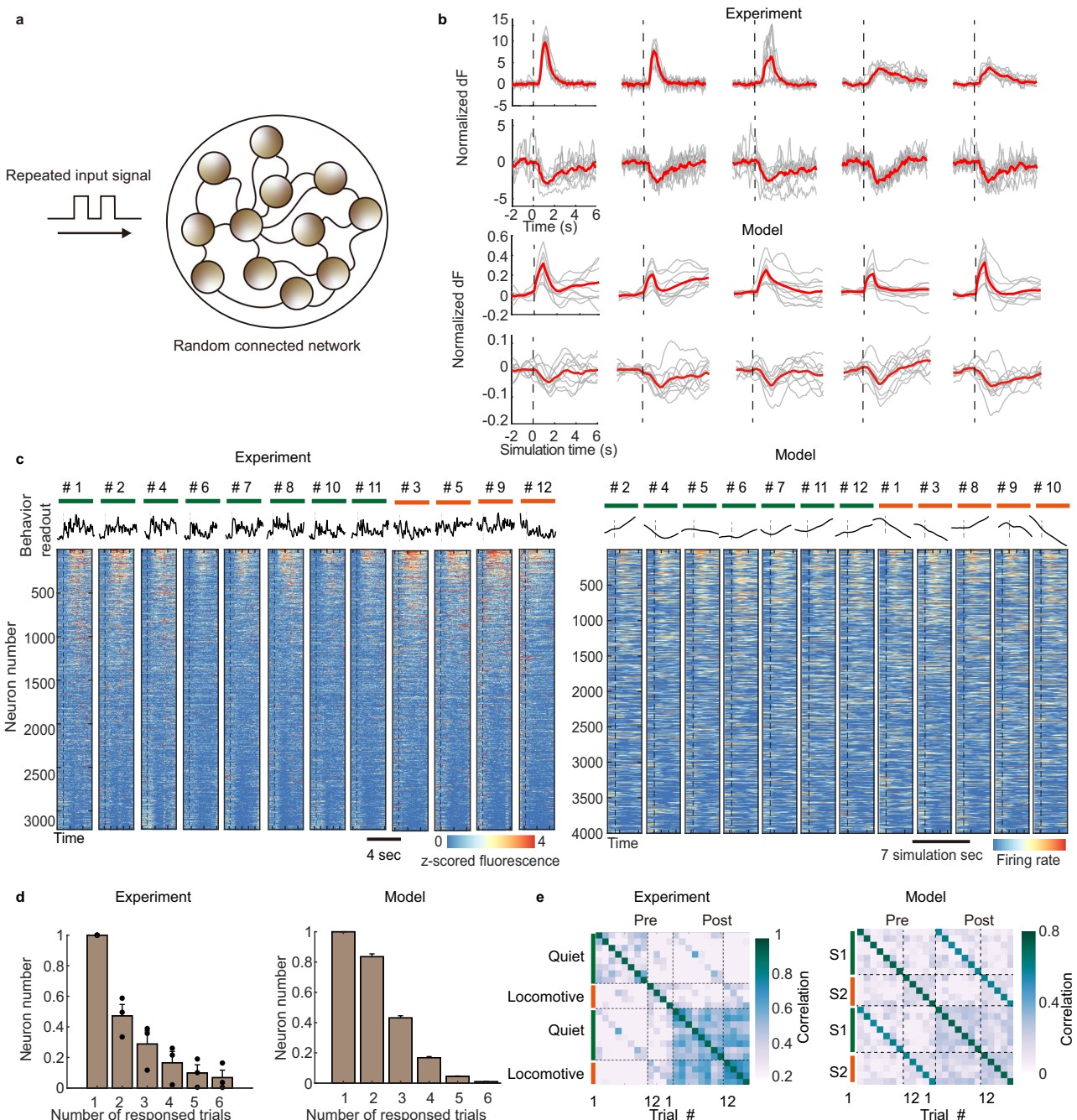

**Fig. 3 | A randomly connected spiking model explains the stimulus-dependent trial variability. a** A randomly connected recurrent neural network is built to simulate the generation of trial variance. Each node follows its dynamic function and is connected by a randomly initialized fully connected network. The input is projected to a proportion of neurons which act as the input cell. **b** Single neuron tuning curves of empirical data (upper) and simulated data (bottom), and dash lines mark the onset of stimulus. **c** The responses of a fixed group of neurons under repeated stimulus of the same parameters (bottom). The behavior readout calculated by the motion energy of the animal is also shown (top). Trials are divided into quiet (marked by green bars), and locomotive trials (orange). The model simulation result is shown on the right. **d** Number of neurons above responsive threshold in 1-6 trials out of total repeats, showing both the experiment result and model output. Error bars: SEM. *n* = 3 mice. **e** Correlation of populational activities, trials are divided into two groups based on behavior readout in (**c**). Source data are provided as a Source Data file.

## A steadier point-like attractor revealed by the in silico simulation

We next speculated what other aspects could influence the trial variability. In the design of the model, two parameters stand out: the external stimulus amplitude and the size of the input neuron population. We verified this speculation by increasing the DBS amplitude in the in vivo experiment while keeping the total repetition and stimulation energy within a range tolerable to mice. As expected, when the

input current increases, more neurons across the cortex are tuned (Fig. 4b). The spatial distribution of activated neurons shows a specific pattern in a particular subject and experiment session (Fig. 4a). However, no topology pattern is found across subjects (Supplementary Fig. 7a–c). Trial variability is quantified via population correlation. The inter-trial correlation is strengthened when a higher electrical current is delivered, consistent with the observation in simulated results (Fig. 4c).

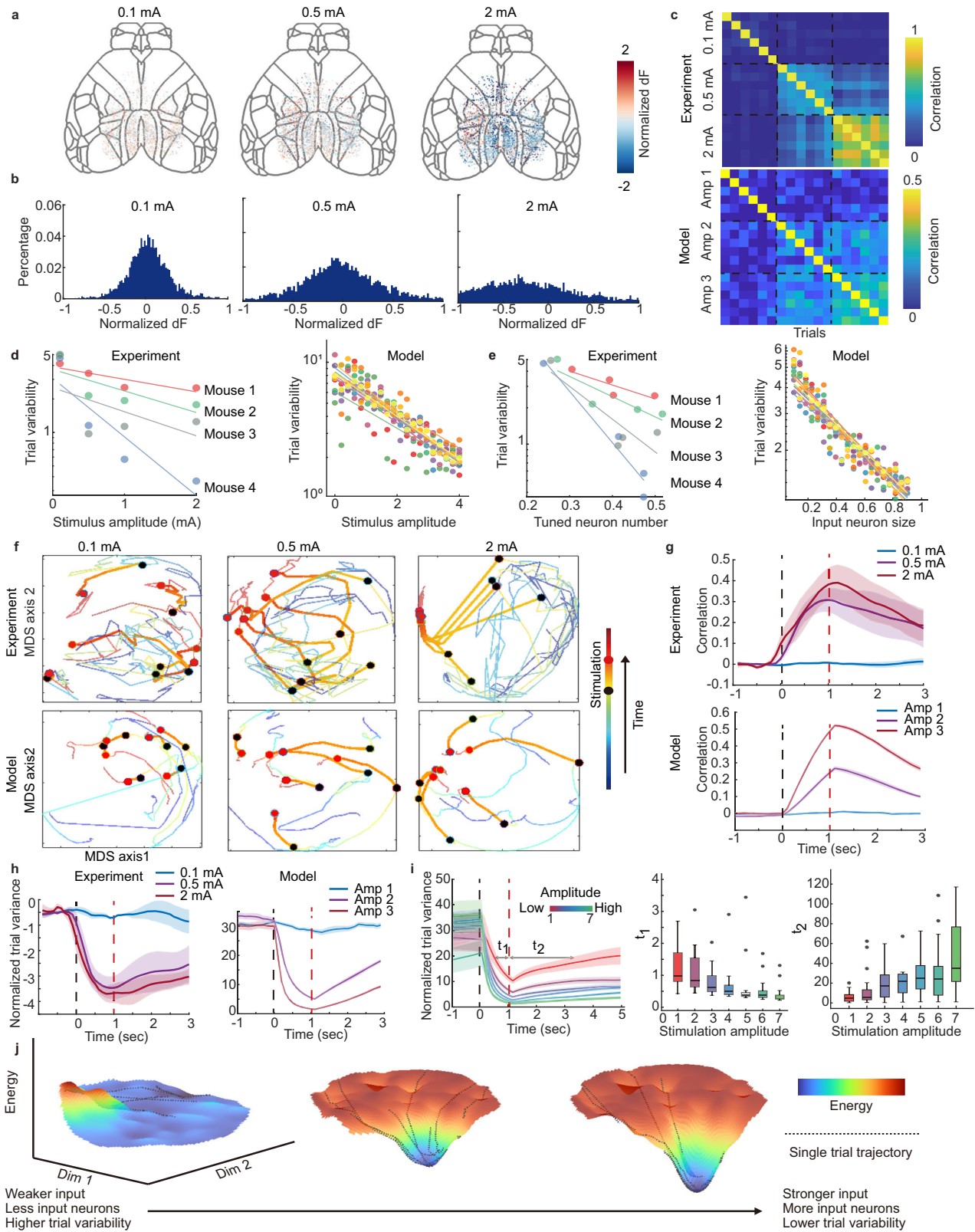

In the simulated network, while changing the input amplitude and the input neuron size, we found a significant relationship between these aspects and the trial variability. The fitting function suggested trial variability follows an approximately exponential decrease function with the growth of these factors (Fig. 4d, e). The property remained robust to the randomized connectivity matrix (Supplementary Fig. 5e–h), suggesting that the exact connectivity of the network

does not contaminate this property. We found similar results in empirical data, the fitting function suggested an approximately linear relationship between the log of trial variability and the DBS amplitude (mean $R^2 = 0.61$ in $N = 4$ subjects) and tuned neuron size (mean $R^2 = 0.80$ in $N = 4$ subjects).

The decrease in trial variability means the formation of a local attractor under the perspective of dynamical systems. To verify this

**Fig. 4 | Increased stimulation shapes a steadier attractor in neural space. a** Trial-averaged cortex neuron response under increased nucleus stimulation.
**b** Histogram of the post-stimuli response of all neurons. **c** The post-stimulation response correlation. Inner- and inter-amplitude correlations are shown in the matrix. **d** Quantification of populational variability changes with the increase of external stimuli amplitude. A linear regression is performed between the stimulation amplitude and the log of trial variance. In model results, each color stands for a random initialization of the connectivity matrix; in empirical results, each color stands for a mouse. **e** Relationship between the populational variability and the input neuron size. In empirical data, we calculated the significantly tuned (the top and bottom 20% of the $z$-scored fluorescence change) neuron proportion as the $x$-axis. Colors have the same meanings as in (**d**). **f** Neural space trajectory visualized on the first and second multidimensional scaling (MDS) components of neural populations. The start and end of stimuli are marked by black and red-filled circles, respectively. Stronger input brings a higher trial consistency and longer traverse in neural space. The upper line shows the empirical results, and the bottom line shows the model-simulated result. **g** The time-dependent correlation change. The start and end of stimulation are marked by black and red dashed lines, respectively. The

empirical result (upper) and the model result (down) are shown. The solid line shows the mean value. The shading represents $\pm$ SEM, $n = 4$ mice. **h** The time-dependent variability of neural population. The start and end of stimulation are shown in the same way as in (**g**). The solid line shows the mean value. The shading represents $\pm$ SEM. $n = 4$ mice. **i** Time-dependent variability with finer-grained amplitude change (left). The time constant of attractor formation (middle) and disappearance (right) with randomized network connectivity (box plotted). The box bounds show the upper and lower quartiles of the distribution. The centered line shows the median. The whiskers show the maximum and minimum. Dots are outliers. $n = 20$ repeated runs with different initialized connections are conducted. **j** Visualization of the energy landscape of the neural system in 3D space. Neural activities are projected into a 2D space in a monotonic manner. The height of the landscape represents the energy, which is approximated by the dynamic velocity of each point in the neural space. Each dotted black line is a neural trajectory of one experiment trial. A deeper energy well emerges when the amplitude of external input or the number of input neurons increases. Source data are provided as a Source Data file.

idea, we visualized the brain state trajectories using multidimensional scaling (Fig. 4f). High-dimensional neural space is projected to a 2D low-dimensional space in a monotonic way. The state space trajectory shows a non-periodic chaotic pattern until the external stimulation is delivered. With the increase of intensity and range of input, the neural state converges to a point-like attractor and forms a radial pattern. To further investigate the dynamic properties of the stimulus-shaped attractor. We calculated the time-dependent trial variability and correlation (Fig. 4g, h and "Methods"). Trial variability is significantly determined by input amplitude (ANOVA test, factor of contrast, $p < 0.0005$, "Method") and time (paired Wilcoxon test, $p < 0.0005$); the correlation analysis reveals the same result ($p < 0.0005$ in ANOVA test of contrast factor and paired Wilcoxon test of time). To understand the dynamic process of trial variability tuned by stimulus, we performed an in silico simulation with stimulations that are finer-grained, higher-repetitional, and extremer in amplitude, making it hard to conduct in vivo. The relaxation time of the variability change is calculated by approximating the time constant of approaching the attractor and then reversing. We found that with a randomized connectivity matrix, the descent time constant decreases with the growth of input amplitude, while the rise time constant increases (Fig. 4i). With the help of the in silico model, we also depicted the energy landscape of the system in densely-sampled neural space. The advantage of visualizing through an energy landscape is that it intuitively depicts the dynamic trend of each point in neural space. The model ensures the energy can be calculated in every corner of neural space, some of which are never visited in experiments. The shape and range of the energy landscape depict the properties of the system under different external regulations. The position of the neural state on the 2-D MDS axis is determined by preserving the cosine distance between each neural state pair in high dimension, and the energy is estimated by calculating the dynamic velocity at each point in neural state space, which reveals the stability of the system[57] (Supplementary Fig. 7d). An energy well emerges when the input intensity and range increases in the model (Fig. 4j). The result suggested that the attractor formed by the stronger stimulus could draw the brain state in a shorter time, and is slower to release the states after the offset of stimulation.

### Imperfectly tuned neurons constitute a dispersed and collaborative coding community

Nucleus stimulation is an arbitrary and extrinsic electrical delivery to cortex neurons. We next investigate if physiologically functional tunings of cortex neurons follow the same rule we found previously. To perform a highly repetitive and reliable sensory stimulation, we presented mice with a drifting grating on a screen positioned in front of the right eye. Gratings of eight different orientations are presented in a

pseudorandom order. Functional imaging of cortex neurons is captured via the RUSH system simultaneously (Fig. 5a). As a non-direct stimulus, the number of tuned neurons is significantly lowered in visual tasks than DBS, and the visualization on multidimensional scaling (MDS) axis no longer captures distinct stimulus tuning effect (Supplementary Fig. 7e, f).

The intensity of sensory input is tuned by changing the contrast of grating bars. Just as what we found in DBS paradigms, time-dependent trial variability of visual tuning also exhibited an apparent decrease at the onset of stimulation, and the attractor is strengthened by sharply contrasted gratings at the population level (ANOVA test, factor of contrast, $p < 0.05$) (Fig. 5b). Dividing the cortex into CCF brain areas reveals a spatial heterogeneity tuning of single-trial variability (Supplementary Fig. 8a). The observation inspires us to calculate the single-neuron tuning variability under the sharpest contrast, searching for a finer-grained spatial map (Fig. 5c). As expected, neurons with reliable tuning properties greatly concentrated in the visual cortex. Surprisingly, we also found highly reliable neurons dispersed at the outer range of the visual cortex. We also calculated the orientation selectivity index (OSI) of neurons in different brain areas, which supports the dispersed distribution of some visual-sensitive neurons outside the visual cortex (Supplementary Fig. 9). The distribution of neural trial variability is calculated in each CCF brain area, and a Wilcoxon rank sum test is performed between each pair of brain areas (Fig. 5d, e). The outcomes indicated a significant difference between VISp, VISpm, and other brain areas, which is consistent with the conventional understanding of functional brain area division.

Decoding accuracy is a direct reflection of trial variability, which is very critical for the applications such as brain–computer interfaces with a limited number of channels for neural recording. A high decoding accuracy normally means a low trial variability for each neuron or neural population. In the contrary, as observed in our in vivo and in silico data, significant trial variability necessitates a sophisticated readout function to ensure stable cognitive output (Supplementary Fig. 10). Trial variability brought us concerns in mesoscale imaging: if neurons are imperfectly tuned, does populational coding facilitate inter-trial coding fidelity[58]? If so, does the coding efficiency constantly increase with the number of recorded neurons? To answer these questions, we calculated the trial-orientation decoding accuracy using neurons located within a certain radius from the center of the visual cortex. A simple KNN classifier was employed as the decoding tool, and the leave-one-trial-out method was used to test the classifiers trained on all other trials. Up to $10^4$ neurons are recorded in a 6-mm diameter cranial window. While spanning the field of view centered at the visual cortex, the decoding accuracy increases at first, but eventually falls before reaching the maximum number of recorded neurons

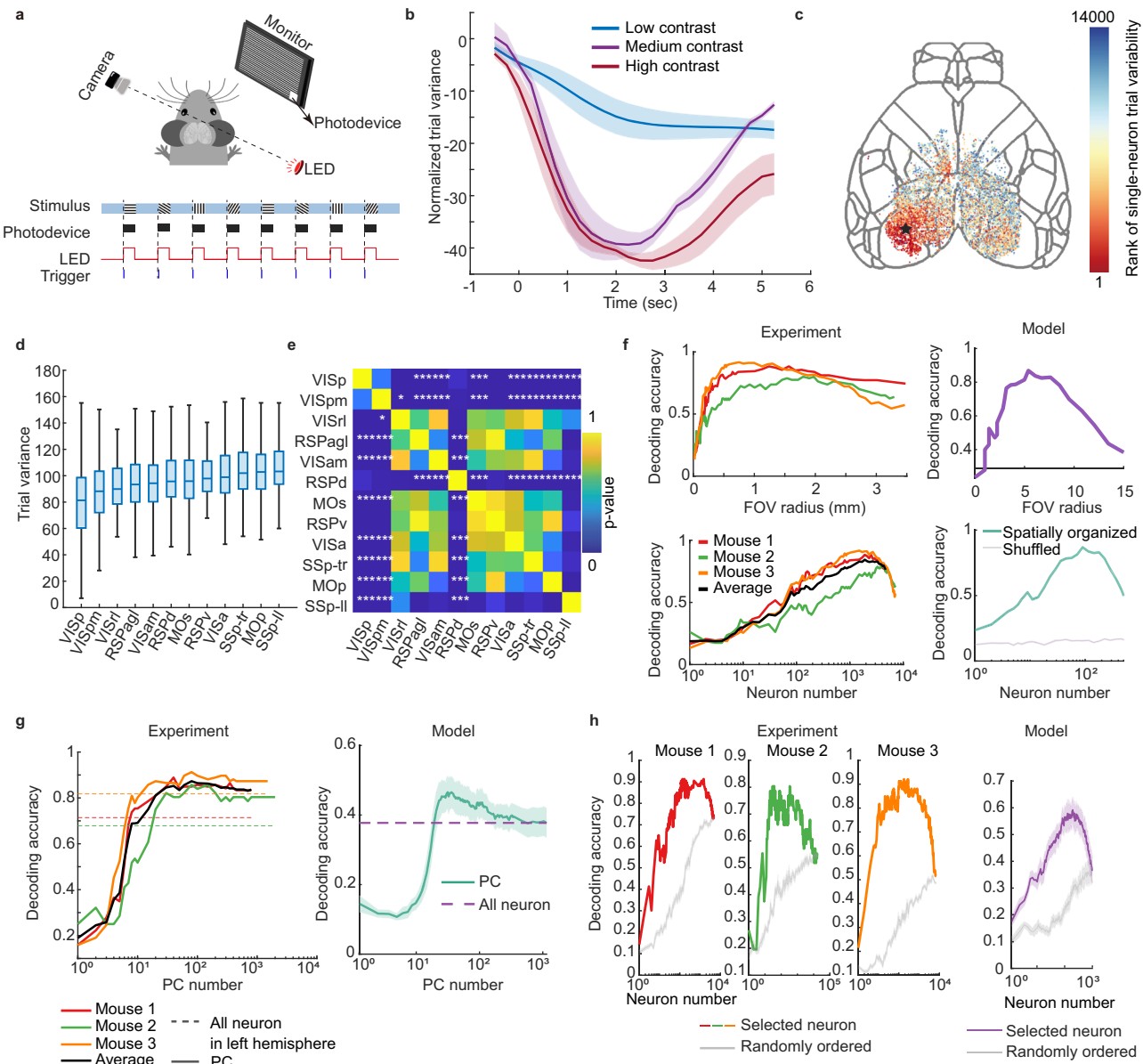

**Fig. 5 | Trial variability and decoding analysis during visual perception. a** A schematic of visual stimulation. Gratings in 8 directions are presented by a screen in a pseudo-random order. **b** Time-dependent mean trial variance during delivery of grating videos with different contrast, shading stands for SEM, $n = 3$ mice. **c** The ranking single-cell trial variance, smaller rank means lower trial-variability. The black pentagram indicates the center of the visual area, which is used as the center of the field of view in (**f**). **d** Trial variance in each brain area, the box indicates the range from the bottom quartile to the upper quartile. The centered line shows the median. The whiskers show the maximum and the minimum of data. $n = 14445$ neurons are included in the total. **e** Two-sided Wilcoxon rank sum test (with Bonferroni correction for multiple comparisons) result between every brain area.

($^*p < 7.6 \times 10^{-4}$, $^{**}p = 7.6 \times 10^{-5}$, $^{***}p < 7.6 \times 10^{-6}$). **f** Decoding grating orientation using increasing neuron numbers (bottom), and the corresponding field of view radius (top). The center of view is shown in (**c**) with a black star. The model result is produced by a setting with spatially ordered functional neurons. **g** Decoding accuracy changes with the number of principal components (PC). The shading represents ± SEM in repeated simulations with different initialized connections. **h** Decoding accuracy changes with the number of neurons. The order of neurons is either random or ranked by coding ability. The shading represents ± SEM in repeated shuffled runs (gray), and repeated simulations (purple). Source data are provided as a Source Data file.

(Fig. 5f). This explains the fact that from a certain stage, spanning the FOV in simple tasks does not result in a better decoding performance. We speculated that this trend is caused by the inclusion of a massive number of irrelevant noisy neurons when the FOV spans to other brain areas. To validate our assumption, we conducted a simulation of a spatially organized network where neurons at the center of view receive most of the projection from the external source, and the surrounding neurons receive indirect input information from their neighbors (Supplementary Fig. 8b). This model successfully reproduced the unimodal accuracy curve we found in empirical data

(Fig. 5f). Given that visual stimulation was the dominant external sensory input in the visual task, we also tried to project the visual cortex neural activities into the principal component axis and calculated their decoding accuracy. The result reveals that by using typically 1–10 percent of total free dimensions, the orientation could be decoded evenly (Fig. 5g) than using all visual area neurons. This is also reproduced by the in silico model. The redundancy of coding dimensions is the direct result of the dimension gap between the number of neurons and the task complexity[59]. Since we successfully accessed neurons spanning far from the visual area, another way of filtering out

trial-consistent information is by screening for reliable neurons in the first place. We managed to find reliable neurons in a separate training time series and added the selected neurons to the pool of the decoding population in order. Like the single-neuron trial variability, the order of selected neurons is not strictly determined by the distance to the center of the visual cortex. This strategy achieved significantly higher decoding accuracy compared with adding the neurons to the decoding pool randomly (Fig. 5h). Also, using a parameterized learning-based recurrent neural network produced an even higher decoding accuracy, meaning there is still room for algorithmic improvement for better performance (Supplementary Fig. 8c, d). These results gave us confidence in brain–computer interface based on mesoscale high-throughput imaging modality. Similar quantitative results were generated using the simulated network.

Altogether, we demonstrated three strategies for decoding the information from highly trial-variant signals, focusing on the targeted FOV, projecting the neural space into lower dimensions, and filtering out the key neurons. Including more neuron units into the scope of observation naturally increases the decoding accuracy at first, but when the FOV grows further into the whole cortex, neuron units must be used wisely to avoid corruption from noise.

## Discussion

Towards the general challenge of trial variability of large-scale neural recording, we found the origin of trial variability from high-dimensional nonlinear connections in self-organized networks. By building a non-linear neural network model, we demonstrated its inevitability through in vivo DBS paradigms and in silico network simulations. With the help of the model, two key factors influencing trial variability were identified, the projection range and the input amplitude, which are proved via both DBS and visual experiments. A physical picture based on the view of system energy helps us understand the attractor's dynamic properties. We also discussed why expanding FOV came with severe trial inconsistency, especially in simple single-sensory tasks, and pointed out several strategies that could facilitate decoding performance. We argue that variability and unrepeatability are natural features of complex systems, by facing instead of disclaiming them, we might understand the brain in a better sense. DBS was selected for our study because it offers precise stimulation in a specific brain region, unlike Transcranial Magnetic Stimulation (TMS). The parameters of DBS can be easily adjusted to meet the specific needs of patients, providing flexibility not as readily available in optogenetics, which is limited in clinical applications due to genetic modification requirements. Furthermore, DBS's effects are reversible, ceasing once stimulation is discontinued. However, DBS is more invasive than TMS, requiring the implantation of electrodes. Unlike optogenetics, DBS does not allow for targeting specific types of neurons. It is crucial to minimize the physical artifacts in neuronal responses caused by electro-stimulation, ensuring responses are due to autonomous neural activity rather than direct electrical interference. Parameters for DBS were controlled and kept within safe limits. To confirm safety, mouse behavior was closely monitored to prevent any discomfort that might arise from the stimulation. Additionally, distinguishing the response delay times of neurons helps verify their autonomous activation. Moreover, the potential limitations associated with electrode implantation need careful consideration. Implantation can lead to acute or chronic tissue damage, including scarring and gliosis at the electrode sites. Such damage can compromise the long-term effectiveness of the stimulation and alter the electrical properties of the tissue, affecting both the distribution and efficacy of the stimulation. Therefore, a microelectrode probe was necessary to reduce the implantation damage.

Single-photon imaging with a parallel read-out strategy is the approach we employed to access large-scale imaging. Despite the effort we made to confront optical aberrations and perform de-scattering and de-background algorithms in the data-processing pipeline, the hemodynamics, nonuniformly labeled neuron clusters, and nonuniform refractive index of brain tissues still induced fatal crosstalk between neurons. Therefore, it is of great importance to develop multi-photon or confocal imaging systems with larger FOV to further increase the fidelity of the neural signals with dense labeling. Otherwise, there are always concerns that imaging artifact contributes to the trial variability we observed. We look forward to the same study being conducted with multi-photon large-FOV imaging.

A straightforward understanding of trial variability is that it reveals the parallel processing property of the nervous system. The neural units of different tasks are most likely multiplexed, and the designed variable constitutes only a part of their total explained variety[60,61]. The emergence of complexly tuned cells through learning saves the processing energy and follows the principle of efficient coding[62]. The highly dynamic and high-dimensional physical world is thus possible to be reflected in the neural space and mental world.

Uncovering information from undesigned variables has been a target in system neuroscience[63,64]. Apart from the supervised mapping of designed task variables such as sensory input[65,66], task performance[67,68], and action features[69,70]; single-trial analysis[71] through the decomposition of high-dimensional neural signals[72–77] are combined with expert knowledge, has revealed evidence of hidden variables embedded under the trial-by-trial variance, such as locomotion[9], arousal[78], attention[79], and learning[80]. Using a Neuropixel dataset recorded by Kenneth's lab[12], we found that behavior performance also caused trial variability during the early cue presentation phase in certain trial types. In the visual discrimination task, mice earned the reward by turning the wheel to indicate which side of the visual cue has higher contrast, or hold the wheel still if the contrast is the same. We found that in trials where contrast difference is high, the trial variability of correct trials is significantly lower compared with wrong trials (Supplementary Fig. 11). This indicated that the visual stimulation is coded with higher fidelity in the brain when the mice are about to make a correct choice. In the future, with mesoscale imaging, we believe more evidence of physical counterparts to the once-considered imperfection of neural coding will be collected. The high dimensional fact of multi-brain area neural signals also raised demands on designing complex behavior tasks, making the neural task complexity comparable to the number of neurons[59]. The growing FOV in mesoscale imaging also calls for training paradigms including multiple sensory input and behavior output, mobilizing cortex-wide neurons. Efforts have been made in designing learning and decision-making tasks on rodents[81–84], and we look forward to better paradigms to be designed in the future.

In terms of decoding, expanding the FOV around the functional brain area to enhance decoding performance is a natural thought for neuroscientists. However, this is not necessarily how the coding community was formed in the brain. Therefore, picking up dispersed coding units from the whole cortex resulted in a better decoding performance than limiting the FOV within a region. We thus suggest a high-precision brain–computer interface might be built using the mesoscale imaging outcomes, only if the information is processed properly and wisely.

## Methods
### Animal and surgeries

**Mouse handling.** All experiments procedures involving the use of live animals and brain tissue were carried out in accordance with the Tsinghua University guidelines and approved by the Institutional Animal Care and Use Committee (IACUC) at Tsinghua University, Beijing, China. Mice were housed in the Laboratory Animal Research Center of Tsinghua University with standard temperature (24 °C), humidity (50%), and a reversed light cycle with lights on from 7:00 am to 7:00 pm).

Mice (cross between Rasgrf2-2A-dCre mice (JAX 022864) and Ai148(TIT2L-GC6f-ICL-tTA2)-D (JAX 030328) expressing the GCaMP6f in a Cre-dependent manner, Jackson Laboratory) were group housed in plastic cages with standard bedding in a room. Both male and female healthy adult mice (20–30 g), aged 8–24 weeks were used in the experiments. Trimethoprim (TMP, 0.25 mg/g, Sigma) was injected into experimental mice intraperitoneally for consecutive two days[44]. Mice received ad libitum food and water before surgery. They were individually housed after surgery for experiments. Recovered mice after craniotomy for imaging and electrode chronic implantation were head-fixed on a commercial passive treadmill with an encoder (labmaker). All animals would be habituated on the head-fixed system at least one week before recording.

**Stimulation electrodes implantation.** Mice were anesthetized with isoflurane (4% induction and 1–2% maintenance, 1 ml/min O2) in the anesthesia machine (R520 IP, RWD Life Science Co.) and injected with meloxicam (5 mg/kg) subcutaneously. The surgical instruments and tables were strictly sterilized at high temperatures and high pressure prior to each surgical procedure. Mice were carefully leveled in a stereotaxic apparatus (68018, RWD Life Science Co.) and their scalp was removed to expose the skull. The temperature is maintained at 37.0 °C with a heating pad (RWD Life Science Co.). After cleaning the underlying bone using a surgical tweezer and blade, a small craniotomy was made over the electrode insertion sites on the left hemisphere.

For the deep brain stimulus experiment, the electrode array with two channels (Pt-Ir, 0.5 MΩ, 30 μm in diameter, shank spacing 150 μm, HK Plexon) was selected as anode and cathode to produce a stimulus current. Before electrode implantation, the shanks of stimulation electrodes were painted with DiI (Invitrogen, dissolved in DMSO, Sigma-Aldrich), allowed to track the electrode pathway. DiI was carefully coated on shanks only to avoid contaminating the tips of electrodes. Then, the stimulation electrodes were implanted into the claustrum (Cla, AP: −0.17 mm; ML: −3.60 mm; DV: −4.35 mm) at an oblique angle of 5° at the insertion speed of 10 μm/s. After electrode implantation, the craniotomy was cleaned and dried with sterile tissue. Then, it was sealed with surgical glue (Vetbond, 3 M) and dental cement was used to fix the electrode in the edge of the skull. After dental cement forms a completed solid layer, the following surgery steps can be carried out.

**Craniotomy operation for wide-field imaging.** Before wide-field imaging surgery, the position of the bregma was recorded by a fixed probe. Then, the probe was removed to reserve operation space for cranial removal. A piece of skull was removed and replaced with a sterile crystal skull. It was cemented with Krazy glue (Elmer's Products Inc). Single neural activities in the whole brain cortex would be recorded by wide-field imaging. Then, a head-post was cemented on the edge of the skull with a layer of dental acrylic. The electrode interface was fixed on the head-post to avoid breaking during mice movement.

Flunixin meglumine (1.25 mg/kg, Sichuan Dingjian Animal Medicine Co.) anti-inflammatory drugs were injected subcutaneously after the surgery for five consecutive days. Following craniotomy surgery, mice are given a recovery period of at least 7 days. Subsequently, we assess the integrity of the craniotomy window and the quality of calcium signals. Only mice that have fully recovered and are healthy are included in the experiments after this recovery period. To monitor the condition of mice post-electrode implantation, we conduct long-term behavioral observations. Mice have been observed to live for over a year post-surgery without showing any behavioral differences when compared to healthy, non-operated mice.

**Histology.** After imaging experiments, Mice were euthanized and perfused intracardially with 25 ml phosphate buffered saline (PBS) followed by 20 ml 4% paraformaldehyde (PFA) solution. The mouse brain is fixed in the 4% PFA solution overnight at 4 °C and sliced into 100 μm thick coronal sections using a vibratome (VT1200 S, Leica) at room temperature.

**The synergetic system of fluorescence imaging and DBS**
**RUSH system.** The RUSH system was described in previous work[43]. In brief, the RUSH system contains 35 sCMOS in a matrix for parallel acquisition of brain-wide calcium events. Briefly, this system is equipped with a customized objective lens, with $1 \times 1.2$ cm$^2$ FOV and 0.35 NA. The image is then acquired by a camera array, which is 14000 in height and 12,000 in width. The overall magnification factor of the optical system is ×8. The fluorescence is excavated at $470 \pm 20$ nm and detected at $525 \pm 20$ nm. The maximum acquisition rate is 30 fps, while we employ 10 fps in most experiments due to the calcium dynamics. An external trigger signal is used to synchronize the start of acquisition of all cameras, and also output to a digital input channel of the electrophysiology recording system.

**Electrical system.** For Cla electrical stimulus, two constant current-controlled stimulators (ISO-Flex, A.M.P.I, Jerusalem, Israel) were connected to deliver biphasic stimulation. The biphasic rectangular stimulus pulse is controlled by the Arduino board with programmable waveforms. A two-channel interface was used to deliver electrical stimulation.

**Visual system.** Visual stimuli were presented on an LCD monitor to the right eye[54]. To adjust the tangent point between the plane of the monitor and the sphere around the center of the right eye was in the center of the monitor, the monitor was fixed at an anticlockwise rotated angle of 30° relative to the anteroposterior axis of the mouse and tilted 20° towards the mouse relative to the gravitational axis. The distance from the center of the monitor to the right eye was 5 cm. The monitor covered 117° of the field of view horizontally and 110° of the field of view vertically.

**Behavior acquisition.** The faces of the mice were lighted using infrared LEDs (850 nm) to acquire infrared facial video during imaging in darkness. A FLIR camera (BFS-U3-51S5M-C, 75fps) was used to record the facial expression and the frame trigger was recorded by a digital input. The wavelength of infrared illumination was selected to detect the facial expression and to avoid the 488 nm wavelength of the RUSH laser. The mice were placed on the passive treadmill to monitor running speed during mesoscale recording. Running speeds were sent back to the electrophysiological Omniplex recording system by analog signal input interface.

**Synchronization between imaging, external stimulation, and behavior devices.** To align the brain imaging, external stimulation, and behavior devices, synchronized signals were designed for each trial, and recorded by digital input or analog input of the Omniplex system. The RUSH camera trigger and facial camera trigger were sent to digital input. To align behavior imaging with DBS, during electrical stimulus, a TTL signal at stimulus frequency was sent back to the digital input interface of the Omniplex system. In visual stimulation, at the start and end time, a small square in the upper right corner of the screen would change from white (start) to black (end) (Fig. 5a). This signal was detected by the photodiode and sent TTL signals to the digital input interface of Omniplex system. In conclusion, all behavior and imaging devices send synchronization signals to the Omniplex system, which aligns them to the same timeline.

**Repeated external stimulation paradigm**
**Electrical stimulus.** For Cla stimulation, all the parameters (100 μs pulse width, 100 Hz in frequency, 1 s in duration) are constant except for the amplitude. Twelve consecutive stimuli (300 μA in amplitude) were delivered to Cla, with a 2–3 min interval between each stimulus

(Fig. 2a). The movement of the mouse was recorded during each trial. In Fig. 4, to induce different levels of stimulation, the stimulation current of 100 μA, 500 μA, and 2 mA were conducted on one mouse six times, respectively.

**Visual stimulus.** Visual stimuli were generated and controlled using the Psychophysics Toolbox in MATLAB[85]. The spatial frequency was 0.1 cycles per degree and the temporal frequency was two cycles per degree. The stimuli were full-screen gratings displayed for 1 s with no interstimulus blank interval of 4 s. To activate population responses under different visual stimulus pattern contrasts, sine-wave drifting grating stimuli were presented at three different contrasts (contrast index: 0.05, 0.5, and 2) and fixed at 45°. Each contrast grating was displayed 6 times in a row (Fig. 5b). In the visual discrimination task, eight kinds of visual stimuli which oriented from 0° to 315° with 45° intervals were used as stimulation materials (Fig. 5c).

**Fluorescence image processing and neuron extraction.** Since the throughput of the RUSH system is quite large, we developed a customized parallel data analysis pipeline based on CNMF-E[47]. The raw video was first registered to remove motion artifacts, then temporally summarized into pixel-neighbor correlation image and peak-noise ratio image. The Hadamard product of both images was used for initializing neuronal candidates. The footprint and temporal activities were further refined through constrained non-negative matrix factorization with a ring model of background. An intensity-based vessel segmentation mask is generated to reject the neurons located on the vessel. In addition, the extracted temporal signals further went through a supervised deep neural network trained by manual inspection to filter out the fake signals resulting from motion artifacts or hemodynamics. After neuronal signals are extracted from raw video, we project neuronal footprints based on cranial windows landmarks to Allen Mouse Brain Common Coordinate Framework version 3 (CCF v3) and achieve cortical area information for each of the neurons.

For widefield activity analysis, we first, down-sampled the original RUSH data 20 times such that each pixel represents about 16 μm in the physical space and aligned the down-sampled capture to CCF v3[86] for further processing. The fluorescence intensity of widefield captures is from in-focus and out-of-focus neuronal signals, and surrounding neuropil signals.

## Simulated neural network model

**Node and network dynamics.** We make minimal prior assumptions on the layer 2/3 neurons, and used the general rate-based random connected network as the basic model[25,55]. The activity variable of each neuron is denoted as $x_i$, $i = 1, 2, \ldots, N$, where $N$ is the number of neurons in the network. A nonlinear response function $\phi(x) = \tanh(x)$ is used as the activation function, determining the interaction strength from neuron to neuron. A connection matrix $J$ determines the connectivity between each neuron pairs, $J_{ij}$ represents the strength of connection from $j$ to $i$. Without further priority, we assume $J_{ij}$ follows a Gaussian distribution, with an average value of 0 and a variance of $g^2/N$. It was proved[87,88] that under these dynamic formulas, the network is able to generate non-trivial spontaneous activities when synapse connections are strong ($g > 1$), and silent otherwise ($g < 1$). We set $g = 1.2$ to 1.3, which leads to a self-organized chaotic network[87]. The dynamic property of a single node is determined by its interaction with its neighbors and its dynamical property:

$$\tau \dot{x}_i = -x_i + \sum_{j=1}^{N} J_{ji} \phi\left(x_j\right) + H_i^{wn} + H_i^{ext} \tag{1}$$

$$\phi(x_i) = \tanh(x_i) \tag{2}$$

$\tau$ is the decay constant of the neural dynamic, which is set as 0.5 s in simulated networks. $H_i^{wn}$ is the low-pass filtered white noise, which stands for the input from unobserved neurons, as well as the unconcerned external stimulation. The dynamic of noise is expressed as:

$$H^{wn}(t) = h_0 \eta(t) + \left[H^{wn}(t-1) - h_0 \eta(t)\right] e^{-\tau_{wn}} \tag{3}$$

$\eta(t)$ is a random variable drawn from standard Gaussian distribution. $\tau_{wn}$ controls the correlation time of the noise and $h_0$ controls the noise amplitude. $H_i^{ext}$ is the intensity of external input.

**External input.** In DBS simulation, the external input is targeted at input neurons selected randomly from the network. The input weight $w_{ki}$ from stimulus $k$ to neuron $i$ is sampled from a half-Gaussian distribution, which is the absolute value of a Gaussian distribution centered at zero, $\bar{w}_{ki} \sim N(0, \sigma_k^2)$, $w_{ki} = |\bar{w}_{ki}|$. So, the average value of the distribution is approximated by the standard deviation of the original Gaussian distribution, $\mathbb{E}(w_{ki}) \sim \sigma$. By tuning the proportion of the input neuron and the variance of distribution, the input range and the input strength could be tuned.

$$H_i^{ext} = \sum_{k=1}^{K} w_{ki} s_k \tag{4}$$

$s_k$ is the stimulus array. In the DBS simulation, we set $K = 1$, $s$ is a square wave input that lasts for 1 s in each trial, and separated by random inter-trial intervals.

In visual stimulation, $K = 8$ different kinds of input are generated, and the input weight and target are randomized individually. So, there might be an overlap between input neuron populations of different orientations, corresponding to the mixed-selectivity neurons observed in vivo.

**Network with spatial pattern.** To simulate the topology pattern found in the functional brain area, we generated a neuron grid with an input center. We assume both the probability of being an input neuron and the input weight assigned to that neuron decreases with the growth of distance $d_i$ from the center. The input neuron is iteratively selected from the neural population with a sampling weight $q_i$ proportional to the inverse of the distance to the center:

$$q_i = 1 - \frac{d_i}{\sum_{i=1}^{N} d_i} \tag{5}$$

The weight of input is:

$$w_{ki} = \exp\left(-d_i^2 / 2\sigma_k^2\right) + \eta_{ki} \tag{6}$$

$\eta_{ki}$ is a small random variable.

**Energy landscape.** To estimate and visualize the energy landscape of the simulated neural space, we first employed multidimensional scaling to reduce the dimension into a 2D space. The distance between neural state pairs is defined as a negative Pearson correlation of neuron state vectors. The energy of each neural space point is approximated by the dynamic velocity defined by the distance between the current state and the next state:

$$v(x(t)) = ||x(t + \Delta t) - x(t)||_2 \cdot x(t + \Delta t) \tag{7}$$

It is calculated by evolving the network state for 1 step based on the dynamic function. The projected neural states do not necessarily cover the whole MDS space, so we further interpolated the energy of the remaining states using linear interpolation. For visualization, the outliers are eliminated to avoid the effects of singular points. And the landscape is smoothed with a 2D Gaussian kernel.

**Behavior readout of simulated network.** To explain the phenomenon that the initial state of the animal influences the stimulus outcome, we simplified the situation by defining the concerned behavior readout as the second principal component of the simulated neural network. This is based on the fact that principal components of neural space often code the general movement of the animal[9]. The initial state of behavior readout is defined as the pre-stimulation activity in a 20-timestep time window. For each trial, the 20 timesteps trace is considered a 20-dimensional feature. We used a k-means classifier to divide the initial states into two groups. The number of clusters is set as $k = 2$, Euclidean distance is used as the distance metric, and the maximum iteration is set as 100.

**Trial variability calculation.** Trial variance is calculated by measuring the variation of the neural responses from the averaged response. The response of $N$ neurons and a $T$ step time window in $j$ th trial could be written as a matrix $R_j \in \mathbb{R}^{N \times T}$, the trial variance is calculated by

$$V = \frac{1}{N_T} \cdot \frac{\sum_{j=1}^{N_T} ||R_j - \overline{R}||_F^2}{||\overline{R}||_F^2} \qquad (8)$$

where $\overline{R}$ is the average of $R_j, j = 1, 2, \ldots N_T$, $N_T$ is the number of trials, $||R||_F = \sqrt{\sum_{i,j} R_{ij}^2}$ is the Euclidean norm of the matrix.

Time-dependent is calculated similarly based on a single-timepoint neural state.

$$V(t) = \frac{1}{N_T} \cdot \frac{\sum_{j=1}^{N_T} ||R_j(t) - \overline{R}(t)||_F^2}{||\overline{R}(t)||_F^2} \qquad (9)$$

For visualization of time-dependent trial variability, the variability trace is normalized by subtracting the baseline $V(t)$ calculated in a pre-stimulus time window, and then smoothed with a Gaussian kernel.

Single neuron variance is evaluated by calculating trial variability of the response of each neuron separately.

### Neural signal analysis

**Signal preprocessing.** We based our analysis on the fluorescence trace resulting from the non-negative matrix factorization. Each neuron trace is detrended to remove the artifact of photon bleaching, and then normalized with a $z$-score method.

**Visual decoding.** Visual decoding was performed using leave-one-trial-out cross-validation. To address the problem of inter-trial variability, each trial is considered as a basic decoding unit, instead of each timepoint. This has increased the difficulty of the decoding task, given the decay component leaks information from neighboring frames. KNN was used as the decoder, with the default $K = 1$ parameter. The population neural vector of a 1 s post-stimulus time window was used as the input to the KNN decoder. Activities from all neurons are concatenated into a feature vector containing $N_{\text{neuron}} \times N_{\text{time}}$ elements in total.

To compare the decoding performance of selected neurons based on single-neuron trial variability and randomly picked neurons, we separate the total trials into two non-overlapped sets, a neuron evaluation set and a decoding set. In the neuron evaluation set, we tested the trial-to-trial variability of each single neuron, and ranked them from low to high. These neurons are then added to the decoding population in order. The decoding accuracy curve is plotted by the populational decoding performance on the decoding set. This has prevented the information from leaking when selecting highly reliable neurons.

### Statistical analysis

To investigate the effect of external stimulus amplitude on trial variability, we grouped the stimulus by the amplitude of stimuli (DBS) or contrast of drifting grating (visual stimulation) and conducted an ANOVA test on the change of trial-variability after the stimulation. Specifically, we first calculated the populational activity in a 1 s pre-stimulus time window and a 1 s post-stimulus time window as a coding populational vector. These populational factors are then used to calculate the pre- and post-stimulation trial deviation from the averaged populational vector, using either Pearson correlation or normalized Euclidean norm distance. The difference of post-and pre-trial variability is considered as a sampling point in the ANOVA test. We consider stimulation type (amplitude or contrast) as the grouping factor. The ANOVA test is conducted by the *anova1* function using MATLAB for $p$ value. To consider the temporal change of trial variability after the onset of stimulation, we use each pre- and post-stimulation trial variability as a sampling point, and grouped them into two groups based on temporal window. A paired one-sided Wilcoxon test is used to evaluate if the difference between the pairs has a non-zero median. The test is conducted using the *signrank* function in MATLAB.

In Fig. 5, to verify the difference of single-neuron level trial variability of different brain areas, we performed a two-sided Wilcoxon rank sum test. Each neuron is considered as a sampling point, and the brain area it comes from is considered as the grouping factor. We conducted the test between each pair of brain areas. The significance test is conducted with *ranksum* function in MATLAB. Considering the multi-comparison case, we conducted a conservative Bonferroni correction and divided the threshold by the times of comparisons $N = 66$. The threshold we used is $^*7.6 \times 10^{-4}$; $^{**}7.6 \times 10^{-5}$; $^{***}7.6 \times 10^{-6}$.

### Analysis of behavior data

**Pupil processing.** The region of the pupil was defined using DeepLabCut to compute pupil area. Continuous video recordings were acquired using a camera mounted above the arena and were processed post hoc using DeepLabCut. First, some frames of the video were handcrafted to label four points on the boundary of the pupil. Second, DeepLabCut labeled the whole video and output the coordinates of the points. Third, the diameter and area of the pupil were calculated based on the coordinates. The diameter was defined as:

$$\left| \min_d \sum_i \left( d_i^2 - d^2 \right)^2 \right| \qquad (10)$$

Where $d_i$ was the distance between each point on the border and the center of the pupil, and d was the diameter of the pupil.

We used the function lsqnonlin in MATLAB to fit the center and the diameter of the pupil. As the four points were labeled nearly on the major and minor axes of the ellipse of the pupil, the area was approximately calculated using the distances of the two pairs of points.

**Motion energy.** We considered a simple pixel-level motion energy. The motion energy $M_t$ of a given frame is approximated by calculating the pixel-wise difference between the current frame $F_t$ and the next frame.

$$M_t = ||F_{t+\Delta t} - F_t||_F \qquad (11)$$

where $|| \cdot ||_F$ is the Euclidean norm.

### Reporting summary

Further information on research design is available in the Nature Portfolio Reporting Summary linked to this article.

## Data availability

The calcium imaging raw data supporting the findings of this study exceeds 30TB in size. To facilitate access, representative demo data used for the analyses is provided in the 'Code Availability' section of the manuscript. The raw dataset is available from the corresponding authors without restriction. Source data are provided with this paper.

## Code availability

MATLAB code for DBS data analysis, visual data analysis, and RNN model simulation from this work is freely available online at https://github.com/Cai-yy/Single-trial-variability.git.

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

## Acknowledgements

This work was supported by the National Key Research and Development Program of China (2023YFC3402600 (G.X.)) and the National Natural Science Foundation of China (62171254 (G.X.), 62088102 (Q.D.)).

## Author contributions

This project was the result of a close collaboration. G. Xiao, Y. Cai, J. Wu, and Q. Dai conceived the idea of this project. G. Xiao designed and developed the multi-mode recording system, with the assistance of H. Xie. G. Xiao developed a surgical process and performed all the mice experiments, with the assistance of J. Xie. Y. Cai developed the stimulus-shaped attractor model, and compared data between in vivo and in silico. Y. Zhang established the pipeline for calcium signal extraction. G. Xiao, L. Wu, and J. Xie performed the preprocessing of experimental data. Q. Dai and J. Wu supervised all aspects of the project. G. Xiao and Y. Cai made figures and wrote the manuscript with feedback from all the authors. J. Wu revised and polished the paper.

## Competing interests

The authors declare no competing interests.
