## [Transparent Peer Review file · Nature Communications]

Mesoscale neuronal granular trial variability in vivo illustrated by nonlinear recurrent network in silico

Corresponding Author: Professor Qionghai Dai

Version 0:

Reviewer comments:

Reviewer #1

(Remarks to the Author)

Xiao and colleagues have submitted an interesting paper that employs the powerful RUSH imaging system to conduct large-scale imaging of the mouse cortex at single-neuron resolution. The study monitors calcium responses to deep brain stimulation (DBS) and analyzes population responses to study trial variability. Additionally, a recurrent neural network (RNN) is constructed to simulate neuronal responses.

While exploring the sources of trial variability is a compelling topic and combining RNNs is popular, there are significant areas for improvement in both presentation and content. Major experimental gaps in the data fail to identify the biological substrates generating the trial variabilities. Without clear data and findings addressing these gaps, the paper seems rushed and does not meet the publication standards of Nature Communications. Overall, it lacks the complete conceptual advance required.

Major concerns

1. The biggest problem with the paper is that the data and results do not convincingly align with the study's topic. Although the authors suggest that "a randomly connected spiking model explains stimulus-dependent trial variability" in Figure 3, there is no clear description or evidence demonstrating this explanation. The traces from the model look different from the experimental calcium traces, with merged baseline sessions and differences primarily emerging after stimuli. The modeled traces appear curved before stimulation, and the separation between trials seems like artifacts from trend elongation before stimulation and forced by the offsets at stimulus onset. These variabilities do not seem related, which is unsurprising as the RNN model lacks critical biological details necessary for generating variability and other computational advantages. The landscape analysis reflects differences between stimulus intensities but not the sources of variability. Larger inputs recruit more neurons and simply cause larger responses, which can be visualized through dimension reduction analysis in neural space visualization and energy landscape figures. None of these results can be considered explanations for trial variability.

2. A second major concern is that the data quality does not match the study's topic. The variability observed in Figure 1b might be due to stimulus drift, such as synaptic vesicle depletion, rather than those more unpredictable trial variability in complex networks. The 'deactivation' shown in Extended Data Figure 4d is suspicious and might be due to focus plane drifting rather than actual inhibition. Because calcium responses mainly reflect suprathreshold firing, where only positive peaks reflect firing, and inhibition brings activity to a flat, low-amplitude state. The negative traces should not mirror positive traces, and the symmetric waveform suggests motion artifacts causing negative values. In Extended Data Figure 8e, where Mouse 3 deviates significantly from the other two. This large difference is ignored and unexplained.

Minor concerns

1. Many analyses appear trivial and unrelated to the study's main question. For example, the improvement in decoding grating orientation with increased neuron numbers is obvious but unrelated to trial variability. Similarly, the decrease in decoding accuracy with increased FOV is expected due to brain size constraints and functional domain boundaries.

2. The manuscript is poorly written, and the figures are carelessly prepared, leading to confusion. Specific examples include:
a. Figure 2a is described as having a pseudo-random variation spanning several minutes in the text (Line 202) but is labeled

- as 3 minutes in the figure.
- b. Figure 2g does not explain the meaning of SN and WF.
 - c. Figure 5f has the top and bottom panels reversed.
 - d. MDS (multidimensional scaling) is used multiple times but never explained.
 - e. Lines 151: "GCaMP6f" should be used instead of "GcaMP6f."
 - f. Lines 257: A period is missing at the end.
 - g. Lines 565: "MATLAB" should be used instead of "matlab."
 - h. Lines 671: "artifact" should be used instead of "artefact."
 - i. Lines 1028: "and locomotive" should be used instead of "sand locomotive."

Reviewer #2

(Remarks to the Author)

The present study investigates the trial variability across multiple brain areas and proved the existence of ineluctable trial variability both in vivo and in silico. Several issues need clarification or improvement.

1. I wonder why the authors choose the visual area as the region of interest. I suggest more literature review on the selection of brain areas in Introduction.
2. The authors should also provide more details on DBS. As we all know, DBS can be used to treat Parkinson's disease, indicating the potential influence to brain. Therefore, it is important to consider the influence of DBS when interpreting the results.
3. Electro-stimulation is consistent with neuronal response, how to rule out that it is not the physical image directly in the body caused by the electro-stimulation, but is caused by the activation of neurons by the neuroelectric stimulation, which triggers the autonomous response of neurons?
4. How to select parameters to study variability in the model construction process, and is it possible that other feasibility factors are ignored during the model construction process?
5. How is noise defined, and could the defined noise be a certain variability factor? Does expanding the FOV introduce more noise, leading to greater variability?
6. Is it possible to explore the impact of variability factors on the behavioral performance of mice? Can this model be used for human brain research?
7. The lack of introduction to statistical analysis methods makes it uncertain whether the results are reliable. Please give more details on statistical analysis, including the 2-way ANOVA in Line 311, the Wilcoxon rank sum test in Line 351. The Pairwise Wilkson test in Line 1069.
8. In Figure 5e, the authors used * to indicate $p < 0.5$. Generally, we used 0.05 as the threshold. In addition, I wonder where the authors considered corrections of multiple comparisons.
9. Please give more details on the clustering in Line 652, including the samples, features, number of iterations, clustering distance.
10. Please check the definition of Euclidean norm of the matrix in Line 659.
11. Please specify the version of "matlab" in Line 565 and "MATLAB" in Line 703. Also, please keep consistent in the manuscript.
12. Please clearly define and describe the variable Hext in Line 606.
13. In Line 512, the meaning of "as describes18" is not clear. Please correct. In addition, please check the "1 × 1.2cm²" in Line 513, the "30°relative" "20°towards" in Line 529.

Reviewer #3

(Remarks to the Author)

This is a very interesting paper which offers an engaging exploration of mesoscale neuronal granular trial variability in vivo, and complemented by simulations using a nonlinear recurrent network in silico. The authors have conducted extensive experiments on mouse brains and provided robust modeling validation. The manuscript is well-organized, and the results are effectively presented.

The study utilizes in vivo imaging through the previously developed RUSH system, which records a large number of neurons at micrometer resolution. The authors have established a comprehensive pipeline to manage the substantial neuronal data collected across more than 12 cortical regions. They designed both electrical and visual stimuli to investigate trial variability at the single-neuron level across the entire dorsal cortex. This study is presented with clarity and efficiency. Overall, this paper achieves a significant milestone that could greatly influence future studies on trial variability. To strengthen the paper further, I have a few minor suggestions:

1. Details regarding the post-surgical recovery of the mice are needed. How long after the craniotomy and the implantation of electrical electrodes do the experiments begin? Additionally, it would be helpful to know the lifespan of the mice following electrode implantation and whether the surgeries impact the mice's health or behavior.
2. The claustrum (Cla) nucleus was chosen as the stimulation site. Many researches have studied this nucleus using some in vitro methods. What are the specific reasons for the author to target this nucleus?
3. The authors tried the electrical stimulus and visual stimulus paradigm to test mesoscale trial variability. What is the difference of electrical stimulus and visual stimulus in trial variability studies? Are there any different characters due to the different kinds of input?

4. The author has combined mesoscale fluorescence imaging and deep brain stimulus as the protocol for investigating trial variability. However, trial variability could be explained and depicted in a local circuit and using repeated sensory input, using merely conventional electrophysiology or 2-photon microscopy. What is the advantage of using mesoscale imaging, and why the authors chose DBS as the stimulus method in this study?
5. In Fig. 5, a very simple KNN model is used as the decoder to decode the grating orientation. Why is KNN chosen as the decoder when there are better coding strategies?
6. Trial variability has long captivated scientists, particularly in the life sciences, where averaging has traditionally been employed to manage it. However, advancements in observational technology promise to significantly transform our analytical approaches. Given these developments, how can we more effectively leverage trial variability in future research?
7. In the manuscript, there are inconsistencies in the spacing between "Fig." and the number. Please ensure uniformity.

Reviewer #4

(Remarks to the Author)

In the manuscript titled "Mesoscale neuronal granular trial variability in vivo illustrated by nonlinear recurrent network in silico", the authors investigated trial variability in large-scale neural recordings by combining mesoscale fluorescence imaging and deep brain stimulation (DBS) in mice. They compared the empirical findings with a simulated neural network model to understand the origin and dynamics of trial variability. The study revealed that trial variability is an intrinsic property of complex, nonlinear neural networks and is influenced by factors such as input amplitude, projection range, and the initial state of the animal. The authors demonstrated that DBS shapes neural dynamics, leading to the formation of a stimulus-dependent attractor state, and that imperfectly tuned neurons can form a spatially heterogeneous coding community, with reliable coding achieved through population dynamics.

To conduct this research, the authors developed an experimental setup that combines their previously developed mesoscale fluorescence imaging (RUSH) and DBS, enabling simultaneous stimulation and recording of large-scale neural activity at single-neuron resolution. This synergistic system allows for a comprehensive study of neural dynamics under various stimulation conditions. The authors also visualized the energy landscape of the neural system under different stimulation conditions using the simulated network, offering a new perspective on neural dynamics. This approach helps to understand the formation and dissolution of attractor states in response to external perturbations. Furthermore, the authors proposed strategies for improving decoding performance in the presence of trial variability, such as focusing on targeted regions, dimensionality reduction, and identifying reliable coding units. These strategies offer valuable insights for future research in systems neuroscience and the development of brain-computer interfaces. The manuscript is of high quality and is suitable for publication in Nature Communications

I only have minor comments for the authors to improve the manuscript: 1. The authors could consider discussing the advantages and limitations of DBS compared to other stimulation methods, such as optogenetics or transcranial magnetic stimulation, in the context of studying trial variability and neural dynamics. This comparison would highlight the specific benefits and drawbacks of using DBS for investigating trial variability and neural dynamics. 2. While the authors focus on the claustrum and its projections to the cortex, it would be informative to discuss whether the findings on trial variability and the effectiveness of the proposed decoding strategies are expected to generalize to other brain regions and neural circuits. 3. What is the computational efficiency of the proposed decoding strategies, particularly when dealing with large-scale neural recordings. The authors can suggest potential optimizations for real-time applications.

Grammar:

1. Line 294, change "trial variability follows an approximate exponential decrease function" to "trial variability follows an approximately exponential decrease function".
2. Line 220, change "The distribution of the z-scored neural fluorescence intensity changes is centered at zeros" to "The distribution of the z-scored neural fluorescence intensity changes is centered at zero".
3. Add missing articles (a/an/the) throughout the manuscript.
4. Use either "in silico" or "in-silico" consistently throughout the manuscript.

Version 1:

Reviewer comments:

Reviewer #2

(Remarks to the Author)

The authors have been very responsive to my prior concerns and addressed them accordingly. I also thank the authors for their collegial response letter.

Reviewer #3

(Remarks to the Author)

The authors have fully addressed my concerns and comments. I agree with the publication of this paper now.

Reviewer #4

(Remarks to the Author)

The authors have addressed my comments and concerns. I recommend its publication in NC.

Manuscript: NCOMMS-24-24504-T

Title: Mesoscale neuronal granular trial variability *in vivo* illustrated by nonlinear recurrent network *in silico*

Aug 22, 2024

We sincerely appreciate the thoughtful comments and constructive suggestions from the reviewers, which have significantly enhanced the quality of our manuscript. We have prepared a detailed point-by-point response letter, incorporating results from additional experiments and corresponding revisions to the manuscript. The original reviewers' comments are presented in black, while our responses are highlighted in blue for clear presentation. Before addressing each specific comment from the reviewers, we provide a brief overview of the major revisions we have made:

- 1) Additional DBS experiments and refined analysis strategies were conducted to validate our findings. We utilized a mesoscale light-field imaging system, which provides high speed 3D imaging across a 3D volume of about $4\text{mm} \times 6\text{mm} \times 400 \mu\text{m}$ [1], to validate our previous results. This approach effectively mitigated potential motion artifacts caused by focal shifts in the previous RUSH 2D system. We reanalyzed both excited and inhibited neurons and revised the trace baseline alignment strategy to better assess the deactivated neuronal traces.
- 2) Further experiments using visual stimuli were carried out to enhance the robustness of our data, considering variability among mice. We included three additional mice in experiments where drifting patterns were presented as visual stimuli. The coding accuracy of neurons was analyzed repeatedly to strengthen our results. Additionally, visual stimuli involving three different contrast levels of drifting patterns were employed to solidify the findings.
- 3) Additional sections of the manuscript were added to provide more comprehensive model details and study background. Information on model establishment and parameter selection has been added to the main text and the response letter. The background discussion now includes comparisons of various modulation methods such as DBS, optogenetics, and TMS. Reasons for selecting and comparing DBS and visual stimuli are elaborated upon. Finally, we meticulously reviewed and corrected all typographical errors in the manuscript.

Point by point responses to reviewers' comments:

Reviewer #1:

Comments: Xiao and colleagues have submitted an interesting paper that employs the powerful RUSH imaging system to conduct large-scale imaging of the mouse cortex at single-neuron resolution. The study monitors calcium responses to deep brain stimulation (DBS) and analyzes population responses to study trial variability. Additionally, a recurrent neural network (RNN) is constructed to simulate neuronal responses. While exploring the sources of trial variability is a compelling topic and combining RNNs is popular, there are significant areas for improvement in both presentation and content. Major experimental gaps in the data fail to identify the biological substrates generating the trial variabilities. Without clear data and findings addressing these gaps, the paper seems rushed and does not meet the publication standards of Nature Communications. Overall, it lacks the complete conceptual advance required.

Response: We sincerely appreciate the reviewer for the careful review and professional comments on our manuscript. In response to the valuable feedback provided, we have made efforts on polishing the manuscript to enhance its presentation. Specifically, we conducted additional experiments using new mice and 3D imaging system to improve data fidelity and address the gaps mentioned by the reviewer. We also carefully re-examined our experimental data, applying more rigorous motion-artifact corrections and enhancing the accuracy of false positive neuron detection. Furthermore, we optimized our model to ensure the consistency between the empirical results and the model outcome. Additionally, we re-organized our explanation of how self-organized systems contribute to trial variability for better presentation of the findings. Detailed responses to each comment are listed below.

Q1: The biggest problem with the paper is that the data and results do not convincingly align with the study's topic. Although the authors suggest that “a randomly connected spiking model explains stimulus-dependent trial variability” in Figure 3, there is no clear description or evidence demonstrating this explanation. The traces from the model look different from the experimental calcium traces, with merged baseline sessions and differences primarily emerging after stimuli. The modeled traces appear curved before stimulation, and the separation between trials seems like artifacts from trend elongation before stimulation and forced by the offsets at stimulus onset.

These variabilities do not seem related, which is unsurprising as the RNN model lacks critical biological details necessary for generating variability and other computational advantages. The landscape analysis reflects differences between stimulus intensities but not the sources of variability. Larger inputs recruit more neurons and simply cause larger responses, which can be visualized through dimension reduction analysis in neural space visualization and energy landscape figures. None of these results can be considered explanations for trial variability.

Response: We thank the reviewer for the insightful comment. For the first concern about the traces from the model, we are sorry about the misunderstanding induced by the previous presentation of the modeled traces with our force of onset alignment in Figure 3. To more accurately present the stimulus-triggered activation and deactivation of neural response, we have adjusted our baseline alignment strategy from setting the zero-moment activity to zero to normalizing pre-stimulus activity to zero. We also slightly increase the input intensity used in the model of Figure 3 to enhance the visualization of the stimulus effect. Based on these results, we can find

that the activation and deactivation shown in Figure 3 are indeed triggered by external stimulation rather than an artifact from elongated trend of baseline drift. Additionally, we show the response of example deactivated neurons trial by trial with raw unnormalized trace, where the deactivation effect could also be better identified (Figure. R1a). In some neurons and trials, such as Neuron #3 Trial #1 and Neuron #4 Trial #6, neural activity shows an increasing trend before stimulus, but decreases upon stimulation onset. The deactivation of neurons by external electrical stimulation may be strange at first glance, but the phenomenon is observed both *in vivo* and *in silico*. A potential explanation is a network mechanism where an activated first-layer neuron transmits its positive activity to the next layer through a negative synaptic connection of an interneuron, leading to the deactivation of downstream neurons. We have updated the figures in the main text and added corresponding discussion.

For the concern of RNN model, it is undeniable RNN is a simplified model of the cortex network, where great number of biological details such as ion channel dynamics, stochastics of spiking generation and synaptic plasticity are not comprehensively described. However, we do believe it is a qualified model to describe the trial-variability we observed, influenced by external input and internal state of animal. This model effectively describes the essential property of unpredictability of high-dimensional nonlinear dynamic system. This kind of continuous activity-based network was proved[2, 3] to be able to generate non-trivial spontaneous activities when synapses connections are strong ($g>1$), and silent otherwise ($g<1$). We used $g=1.2$ in our model, which enabled our network to show complex and diversiform patterns even without external input (Figure. R1b). This kind of model is widely used to reproduce multiple computational process in brain network of different species. Such as the brain state change in zebrafish[4], the context-dependent decision making process in macaque monkey prefrontal cortex[5], and the sequence generation and memory mechanism[6]. Therefore, despite its simplicity, we contend that the RNN is capable of generating variable complex patterns and performing cognitive computational tasks, making it a suitable model to describe the self-organized variability of interest in our study. Also, we intend to make the model simple, which is beneficial to find the sufficient origin of trial variability without disturbance of complicated parameters.

It is true that the landscape is a visualization tool of energy distribution along the neural space, while the true origin of variability lies in the generative model of recurrent network. The advantage of using an energy landscape is that it intuitively depicts the dynamic trend of each point in neural space. And the model ensures the energy could be calculated in every corner of neural space, some of which are never visited in experiments. However, neither the position (in multidimensional scaling (MDS) axis 1 and MDS axis 2) or height (energy) of the energy landscape is directly related to response magnitude. Rather, the position of each neural state along

2 MDS axis is determined by cosine distance $D(x_1, x_2) = 1 - \left(\frac{x_1}{\|x_1\|} \right)^T \left(\frac{x_2}{\|x_2\|} \right)$ between each neural state pairs

in high-dimensional neural space, where x_1 and x_2 are 2 neural population vectors. It could be seen from the definition that this distance metric is magnitude-invariant, because each populational vector is divided by its modulus-length. The energy axis is estimated by calculating the dynamic velocity at each point in neural state space[7], which is also not directly related to the magnitude of response or the number of the responsive neurons. Therefore, the energy well emerged after stimulation is not a depiction of stronger neural response, but

a depiction of more similar neural population vectors (measured by cosine distance) and more stable system state (measured by the dynamic velocity in neural space). Therefore, the landscape visualizes the depression of trial variability with increased input intensity throughout a wide range of neural space. We give a simple example (Figure. R1c) to better present the idea. In this case, a 4-dimensional neural space at 5 distinct time points is shown. Note that although t_1 , t_3 and t_4 recruits more neurons, they end up far from each other in MDS axis, because they recruit distinct groups of neurons. And although t_5 and t_4 has distinct response magnitude, because their close populational correlation, the energy is low at these 2 states, which means the system reaches a more stable state at this timepoint. We have added corresponding illustrations in the main text for better clarification.

Figure R1 | Simulated results of RNN model. (a) Example neurons which are deactivated by external stimulation. Both Trial averaged and trial-by-trial responses are shown. (b) Spontaneous activity of simulated network without external input shows non-trivial patterns and rich dynamics. (c) An example 4-D neural space at 5 timepoints (top), the MDS dimension reduction result (bottom left), and the trajectory in the MDS-energy space (bottom right).

Q2: A second major concern is that the data quality does not match the study's topic. The variability observed in Figure 1b might be due to stimulus drift, such as synaptic vesicle depletion, rather than those more unpredictable trial variability in complex networks.

The 'deactivation' shown in Extended Data Figure 4d is suspicious and might be due to focus plane drifting rather than actual inhibition. Because calcium responses mainly reflect suprathreshold firing, where only positive peaks reflect firing, and inhibition brings activity to a flat, low-amplitude state. The negative traces should not mirror positive traces, and the symmetric waveform suggests motion artifacts causing negative values. In Extended Data Figure 8e, where Mouse 3 deviates significantly from the other two. This large difference is ignored and unexplained.

Response: We thank the reviewer for the comment about data quality. To address this concern, we have repeated our experiments with enhanced data quality by using a 3D mesoscale light-field microscopy [1], 3D motion correction, and adding the number of mice subjects.

For the first concern about potential stimulus drift, we cannot deny that the long-time drift of the stimulus will bring a certain degree of trial variability. During multiple stimulus, it is true that synaptic vesicle depletion will cause stimulus drift, resulting in trial variability. Vesicles are located at different pools, including readily releasable pool (RRP), recycling pool and reserve pool [8]. The rates of endocytosis compared to exocytosis determine how quickly the available synaptic vesicle pool is depleted, in turn influencing presynaptic efficacy. A brief high-frequency stimulus (HFS, 100 Hz, 40 pulses) has been shown to deplete primarily the readily releasable pool (RRP) of transmitters. This stimulus protocol is too fast to affect the reserve and/or resting pool of vesicles, and the HFS (100 Hz, 100 pulses) did not depress the synaptic response completely because the emptied RRP is continuously restocked with fresh vesicles[9]. Rapid trains of action potentials cause the synapse to expend the releasable vesicles more rapidly than they are replaced—the replacement process is termed refilling—and so such episodes of activity eventually deplete the pool [10-12]. Since the exhausted pool cannot provide vesicles for exocytosis, a form of short-term synaptic depression, termed depletion. During periods of rest, the synapse recovers from depression as the pool refills. After being emptied, the RRP at hippocampal excitatory synapses refills completely in <30 s [10, 12, 13]. Moreover, the interval time between two trials is several minutes, which is enough for synaptic vesicle recovery. On the other hand, variability will show a specific trend as stimulus drift over periods of days to weeks, such as odor-evoked[14] responses, visual stimulus evoked response[15] and memory enhancement[16], which have all been carefully exploited before. It is difficult to distinguish significant stimulus drift over a very short period of time.

For the potential motion artifacts, we are sorry for the suspicious deactivated traces in Extended Data Figure 4d. We do agree that 2D imaging may have axial motion artifacts which are hard to be removed. To avoid the concern about motion artifacts, we have repeated the experiments during revision with high-speed 3D imaging by employing our developed 3D light field microscopy[1]. This technique provides snapshot recording of 3D information, enabling accurate 3D motion registration including the z-axis motions, which is not possible with the 2D imaging previously utilized. With scanning light-field microscopy, we achieved about 2 μm lateral resolution and 10 μm axial resolution within a 3D depth range of 400 μm to cover enough depth of layer 2/3 neurons. Moreover, we have utilized a 3D-registration pipeline for light field microscopy where the motion on lateral axis on each viewing angle is corrected, so that after 3D reconstruction, the axial motion artifact gets corrected as well[17]. We believe these upgrades in imaging system and algorithms have greatly improved our data fidelity by correcting axial movement. With our new data, we repeated our previous analysis (Figure R2a, b). We selected several example neurons that were activated and deactivated. We again found these neurons dispersed across the ipsilateral side of electrical stimulation. As suggested by the reviewer, the inhibition brings the neuronal traces to a low amplitude state, which can be observed in both trial-averaged responses and raw temporal traces. Based on these findings, we have updated Extended Data Figure 4d in our manuscript with the new data.

For the individual difference in Extended Data Figure 8e, one contributing factor may have been the initial choice of the axis range for the curves, which we have now adjusted from 0.65–1 to 0-1 in Extended Data Figure

8e to provide a clearer view. Additionally, we considered that other factors could have influenced the observed difference, including the internal state of the animal, the labelling density of GCaMP6f, or the experiment environment. To further consolidate our data, we repeated the experiment with 3 additional mice, employing light field microscopy alongside our 3D motion correction algorithm. The updated results are shown in Figure R2c, and this new data has been incorporated into Extended Data Figure 8e. Despite these variations, our overall conclusion remains the same. With the application of a complex read-out function, neural coding can remain stable despite the inherent trial variability.

In summary, we have enhanced the quality of our data through upgrades to our imaging system, better motion artifact registration, and the inclusion of additional subjects. We believe these improvements during revision have elevated the standard of data presentation. Our main findings remain consistent: specific neurons are deactivated by external electrical stimulation. Moreover, using a nonlinear readout function, we can accurately decode angle information despite the inevitable trial variability.

Figure R2 | Improvement of data quality. (a) Spatial distribution of activated (red) and deactivated (blue) neurons on the cortex showing on the maximum intensity projection (MIP) across 200 μm of the standard deviation of the recorded 3D volumes. (b) Top, Trial-averaged post-stimulation trace (bold), and single trial response (gray). 5 activated and 5 deactivated neurons are shown respectively. Bottom, temporal trace of selected neurons, each red vertical line stands for the start of a stimulation. (c) Repeated experiment of visual decoding with learning-based methods, adding 3 more mice for consolidation.

Q3. Many analyses appear trivial and unrelated to the study's main question. For example, the improvement in decoding grating orientation with increased neuron numbers is obvious but unrelated to trial variability. Similarly, the decrease in decoding accuracy with increased FOV is expected due to brain size constraints and functional domain boundaries.

Response: We sincerely thank the reviewer for the comment. We apologize for any confusion caused by the presentation in our previous manuscript. We have restructured the logical progression between sections to enhance the clarity and coherence of our conceptual ideas.

We believe decoding accuracy is a direct reflection of trial variability, which is very critical for the applications such as brain-computer interfaces with limited number of channels for neural recording. A high decoding accuracy normally means a low trial variability for each neuron or neural populations. In the contrary, as observed in our *in vivo* and *in silico* data, significant trial variability necessitates a sophisticated readout function to ensure stable cognitive output. The discrimination index[18] of optimal linear discrimination strategy could be formulated as:

$$(d')^2 = (r_2 - r_1)^T \left[\frac{1}{2}(C_1 + C_2) \right]^{-1} (r_2 - r_1)$$

where $(r_2 - r_1)$ is the vector difference between the mean ensemble neural response, $\frac{1}{2}(C_1 + C_2)$ is the noise covariance matrix averaged over two stimulation conditions. It could be seen from the formula that higher trial variability leads to higher noise correlation and results in lower discrimination value, which means lower decoding accuracy.

Consider a case where there are 2 kinds of distinct stimulations, each stimulation is repeated for 5 trials (Figure. R3a). When the trial variability is low, trials are well clustered by stimulus type, and could be easily decoded by a linear decoder or nearest neighbor decoder. However, when the trial variability gets higher, the classification boundary with linear classifier becomes vague. When the situation is worse, well-designed nonlinear classification boundary is required in order to decode the stimulus type with high accuracy. We have proved with our *in vivo* experiment and *in silico* simulation that trial variability is inevitable. Therefore, how our cognition of an identical subject remains stable under dynamic changes of coding strategy becomes a non-trivial question.

The decoding ability does not simply rise with the number of neuron number in Figure R4a (Figure 5f in the revised version) and Figure R4b (Figure 5h in the revised version). The statement is true only when neurons contain similar task-information (Figure R4b, gray traces). We agree that the decrease of decoding accuracy with the increase of FOV is related to the functional boundary of visual area. We strengthen the importance of using the correct cortical area and correct neurons while performing decoding tasks in applications such as brain-computer interface. There are some other ways that could decode the stimulus type under high trial variability (Figure. R3b). For instance, by only read from the task-relevant neurons (Figure R4b, colored traces). Ignoring those neurons whose signals are irrelevant to stimulus type, the same decoder achieves better decoding performance in trial-variable data. And also, if the neural response is well projected to a task-relevant axis in Figure R4c (Figure 5g in the revised version), the decoding performance could also be improved. These strategies may be used by the down-stream visual network in brain in order to generate a stable cognition in spite of trial variability. We believe this is a very important practical application for our study of trial variability, which in this case is very related to the main topic of this manuscript.

In conclusion, we think discussion of coding ability is both relative and necessary under the main topic of trial variability in this manuscript. Stable cognitive coding despite significant trial variability is one of the cores of bio-intelligence. We provided some possibility of realizing such coding ability and simulated the result of *in silico* experiments, which agrees with the decoding curves of *in vivo* experiments.

Figure R3 | Relationship between trial variability and coding ability. (a) Diagram of increased trial-variability in neural space. Each dot is a single trial, different color stands for different stimulation type. Dashed lines are classification boundaries of either linear or nonlinear decoding methods. (b) Diagram of decoding with nearest-neighbor read-out function.

Figure R4 | Decoding accuracy analysis during visual stimulus. (a), Decoding grating orientation using increasing neuron numbers (bottom), and the corresponding field of view radius (top). (b), Decoding accuracy changes with the number of neurons. The order of neurons is either random or ranked by coding ability. (c), Decoding accuracy changes with the number of principal components (PC)

Q4. The manuscript is poorly written, and the figures are carelessly prepared, leading to confusion. Specific examples include:

a. Figure 2a is described as having a pseudo-random variation spanning several minutes in the text (Line 202) but is labeled as 3 minutes in the figure.

Response: We thank the reviewer for the suggestion. We are sorry for the confusing expression in the text, since the interval time is set around 3 min but not specific to 3min, to reduce the mouse anticipation and also try to keep the interval time in a similar level. Now we have changed the Figure 2a to make it clear and be consistent with the description.

b. Figure 2g does not explain the meaning of SN and WF.

Response: We thank the reviewer for pointing out. We have added the meaning of SN (single-neuron level) and WF (wide-field level) in the Figure 2 caption and also in the corresponding text.

c. Figure 5f has the top and bottom panels reversed.

Response: We thank the reviewer for pointing out. We have changed the text corresponding to the figure panels in the Figure 5F caption.

d. MDS (multidimensional scaling) is used multiple times but never explained.

Response: We thank the reviewer for pointing out. We have added the meaning of MDS in the corresponding text.

e. Lines 151: "GCaMP6f" should be used instead of "GcaMP6f."

Response: We thank the reviewer for pointing out the problem. We have revised "GCaMP6f" in the text.

f. Lines 257: A period is missing at the end.

Response: We have revised accordingly and added a period at the end.

g. Lines 565: "MATLAB" should be used instead of "matlab."

Response: We have revised "MATLAB" in the text.

h. Lines 671: "artifact" should be used instead of "artefact."

Response: We have revised "artifact" throughout the main texts.

i. Lines 1028: "and locomotive" should be used instead of "sand locomotive."

Response: We thank the reviewer for pointing out. We have revised "and locomotive" in the text. We have also further carefully proofread the manuscript to improve the presentation.

Reviewer #2

The present study investigates the trial variability across multiple brain areas and proved the existence of ineluctable trial variability both in vivo and in silico. Several issues need clarification or improvement.

Response: We sincerely appreciate the reviewer for the careful review and professional comments on our manuscript. Detailed responses to each comment are listed below.

Q1. I wonder why the authors choose the visual area as the region of interest. I suggest more literature review on the selection of brain areas in Introduction.

Response: We thank the reviewer for the suggestion. We have added more literature review on the selection of brain areas in Introduction. We selected the visual area as our region of interest for two primary reasons. First, the visual cortex is among the most extensively mapped and understood areas of the brain, featuring a well-

documented layered structure and distinct functional zones. This detailed mapping facilitates precise experimental manipulation and simplifies the interpretation of results. Second, the visual cortex's location in the occipital lobe at the back of the head makes it relatively more accessible for mesoscale imaging. Additionally, we have expanded our literature review concerning the selection of brain areas, which is now included in the introduction.

“The regions surrounding the primary visual cortex in mice have been linked to various types of visual stimuli, including drifting patterns[19] and natural images[20]. Neuronal ensembles in these areas have been shown to encode visual stimuli effectively[21] and maintain stable over several weeks[22]. Recent studies employing two-photon calcium imaging have demonstrated that the activities of neuronal populations provide a more accurate prediction of visual stimuli. This suggests a robust and durable encoding capacity within these cortical areas[19]”.

Q2. The authors should also provide more details on DBS. As we all know, DBS can be used to treat Parkinson’s disease, indicating the potential influence to brain. Therefore, it is important to consider the influence of DBS when interpreting the results.

Response: We thank the reviewer for the insightful suggestion. We have added specific parameters, neuronal mechanisms, and clinical applications of DBS in the revised manuscript and described the potential clinical influence of DBS on our results.

“To effectively tune neurons expanding multiple functional cortical areas, we started with DBS[23] as the stimulus modality under specific parameters (100 μ s pulse duration, 100 Hz frequency, 1 s stimulation time and 300 μ A amplitude). DBS is a widely accepted technique used to modulate abnormal neural activity by causing depolarization or hyperpolarization of neurons near the electrode[24]. The neuron firing rates would be modified either by enhancing or inhibiting their activity. Additionally, DBS can stimulate the release of various neurotransmitters, such as dopamine[25, 26]. Clinically, DBS is extensively employed to treat the Parkinson’s disease by modulating the abnormal firing patterns in the subthalamic nucleus[27], to reduce symptoms of movement disorders, influencing mouse behavior. DBS can modify the functional connectivity between different brain regions, resulting in either increased synchronization or desynchronization across the network. Chronic DBS has also been shown to induce neural plasticity[16], potentially altering the brain’s response to experimental conditions over time. In our study, mice received DBS on a fresh day to prevent trial variability caused by changes in network connectivity or neural plasticity. Behavior was also well defined to assess its impact on trial variability on our results.”

Q3. Electro-stimulation is consistent with neuronal response, how to rule out that it is not the physical image directly in the body caused by the electro-stimulation, but is caused by the activation of neurons by the neuroelectric stimulation, which triggers the autonomous response of neurons?

Response: We thank the reviewer for the insightful question. First, we recorded cortex-wide calcium neural activities while delivering electrical stimuli to the deep brain nucleus at high speed within several hours. It is important to note that optical and electrical signals represent two distinct modes of signal detection. Second, deep brain stimulation is recognized as an effective clinical treatment for movement disorders. Prior to delivering DBS, we reviewed the literatures to ensure that our parameters were within safe and recommended

limits. 100 Hz was considered as an effective frequency, and the duration 100 us is normally used in clinical and mouse experiments. Additionally, we utilized a minimally invasive stimulus probe (diameter = 35 μm) to prevent excessive total power delivery. Third, we monitored the facial expressions of the mice during experiments to ensure that the stimulation did not cause physical discomfort. Finally, we have also discussed this potential limitation and influence of physical image of the body in the revised discussion.

“It is crucial to minimize the physical artifacts in neuronal responses caused by electro-stimulation, ensuring responses are due to autonomous neural activity rather than direct electrical interference. Parameters for DBS was controlled and kept within safe limits. To confirm safety, mouse behavior was closely monitored to prevent any discomfort that might arise from the stimulation. Additionally, distinguishing the response delay times of neurons helps verify their autonomous activation. Moreover, the potential limitations associated with electrode implantation need careful consideration. Implantation can lead to acute or chronic tissue damage, including scarring and gliosis at the electrode sites. Such damage can compromise the long-term effectiveness of the stimulation and alter the electrical properties of the tissue, affecting both the distribution and efficacy of the stimulation. Therefore, microelectrode probe was necessary to reduce the implantation damage.”

Q4. How to select parameters to study variability in the model construction process, and is it possible that other feasibility factors are ignored during the model construction process?

Response: We thank the reviewer for the insightful question. We have added more details about the parameter selection in the revised version. In this work, we have imposed minimal assumption in our simple RNN model to show the trial variability as a naturally emerging property in the high-dimensional dynamic system. The number of adjustable parameters is kept relatively low. In our model, the main parameters to select are the variance of random connectivity g , the decay constant of calcium dynamics τ , intensity of external input

H_i^{ext} , the number of input neurons N_{in} , and the amplitude of external noise H_i^{wn} .

g is an important parameter which determines the dynamic property of the simulated system. It was proved[2, 3] that under these dynamic formulas, the network is able to generate non-trivial spontaneous activities when synapses connections are strong ($g>1$), and silent otherwise ($g<1$). We used $g=1.2-1.3$ in our model, which enabled our network to show complex and diversiform patterns even without external input, reflecting the internal state of the brain (Figure. R1b).

The decay constant of calcium dynamics τ determines the dynamic change rate of simulated traces. We have chosen $\tau = 0.5$ to best fit the dynamic spiking width of simulated network and the empirical data.

The intensity of external input H_i^{ext} and the number of input neurons N_{in} are recognized as two main factors influencing the trial variability we have studied (Figure. 4d and 4e).

The external noise term H_i^{wn} includes the influence from those unobserved parts to the cortex network, such as the deep nucleus projection, and the light and sound noises due to imperfect experiment environment. The noise term could induce certain unpredictability into the simulated network. But through simulation, we proved that even when decreasing the noise value to zero, our observed trial variability is still preserved. (Extended data Figure. 5b)

We acknowledge the existence of other variables that is ignored during the modelling process, such as the synaptic plasticity, heterogeneity of neural population, and higher order dynamics of calcium traces. Like Geroge E. P.'s famous quote, "Essentially, all models are wrong, but some are useful". The model is overall a simplification of the cortex network, but we believe it has captured the core origin of trial variability that is the high-dimensional non-linear property even without other complicated components involved. And it succeeded in predicting the phenomenon of decrease of trial variability with increased input and the coding ability under distinct stimulations. Therefore, we believe this is a simplified but useful model in the term of explaining trial variability.

Q5. How is noise defined, and could the defined noise be a certain variability factor? Does expanding the FOV introduce more noise, leading to greater variability?

Response: We thank the reviewer for the great question. We have added more details and discussion in the revised version. The noise term H_i^{wn} in the network is expressed as a filtered white-noise[4, 6]

$$H^{wn}(t) = h_0\eta(t) + [H^{wn}(t-1) - h_0\eta(t)]e^{-\tau_{wn}}$$

Where $\eta(t)$ is a random variable drawn from standard Gaussian distribution. τ_{wn} controls the correlation time of the noise and h_0 controls the noise amplitude.

The noise term mainly described the existence of projection input from other brain areas that was unobserved, and also the influence of sensory stimuli from the imperfect experiment environment. It was proved[6] that adding the noise term would improve the robustness of the network if it is trained to fit the empirical data.

The noise is indeed a variable factor that may leads to trial variability. It could be seen from the dynamic system formula $\tau\dot{x}_i = -x_i + \sum_{j=1}^N J_{ji}\phi(x_j) + H_i^{wn} + H_i^{ext}$, that the input to a certain node contains three components.

The first one is the influence of other nodes in the network; the second one is the noise we set, and the third is the external stimulation (Figure. R4a). Altogether, the fluctuation of the three components caused the overall fluctuation of the network.

Unlike the external stimulation, noise intensity does not simply change the level of trial variability (Extended data Fig.5b). This may due to the fact that when the noise is low, the network could easily be trapped in local stable points defined by the network structure, which causes the discrepancy between repeated stimulations. To further clarify the contribution from the noise, we calculated the noise input contribution

$$c_i = \frac{\sum_{t=1}^T H_i^{wn}(t)^2}{\sum_{t=1}^T \left[\sum_{j=1}^N J_{ji}\phi(x_j(t)) \right]^2}. \text{ We plot the trial variability against the averaged } c_i \text{ among all neurons, } \langle c_i \rangle$$

(Figure. R4b). The result shows that when noise is low, the network is less robust to the randomized connectivity

matrix. In our other simulated networks, we kept $\langle c_i \rangle < 1$ to ensure the noise term does not dominate the network activity.

Figure R4 | Analysis of noise term. (a) Example neurons showing the component of network projection, noise and external stimulation, with increased noise level. **(b)** Relationship between the $\langle c_i \rangle$ (noise / network projection) and the trial variability.

Q6. Is it possible to explore the impact of variability factors on the behavioral performance of mice? Can this model be used for human brain research?

Response: We thank the reviewer for the inspiring question. We have verified the hypothesis using a Neuropixel dataset recorded by Kenneth’s lab [28]. In this dataset, the mice were trained to perform a visual discrimination task, and approximately 30,000 neurons are recorded in 10 mice, 39 sessions.

We found that behavior performance indeed caused trial variability during early cue presentation phase in certain trial types. Trial variability as early as before the first move of the animal could predict the trial outcome.

In the visual discrimination task, mice earned the reward by turning the wheel to indicate which side of visual cue has higher contrast, or hold the wheel still if the contrast is the same (Figure. R5a). To evaluate the trial variability during cue presentation period, we calculated the population vector of the recorded neurons after the onset of stimulation and before the first move. We then calculated the correlation between population vectors during different trial repeats. Higher inter-trial correlation means lower trial variability, and vice versa (Figure. R5b). We found that in trials where contrast difference is high, the trial variability of correct trials is significantly lower compared with wrong trials (Figure. R5c). This indicated that the visual stimulation is coded with a higher fidelity in the brain when the mice is about to make a correct choice. We speculate that the cognitive mechanism behind such phenomenon may be various, such as attention, intension of movement or learning-related memory

extraction. Further investigation into this topic shall be an interesting field. We have added corresponding discussion in the manuscript.

Figure R5 | Trial variability related to behavior performance. (a) Diagram of visual discrimination task[28]. The mice earned the reward by turning a wheel to the side where the contrast of grating is higher, or hold still if the contrast is the same. The mice could act as soon as the visual stimulation is presented, we took the time-window from the onset of stimulation to the onset of first move (shaded in green) in our trial variability analysis. ACW, anticlockwise; CW, clockwise. (b) Inter-trial correlation matrix of an example session, with different cue types. Trials are sorted by the performance outcome. (c) Inter-trials correlation grouped by the trial outcome and the cue contrast. Paired Wilcoxon signed rank test is used, *, $p < 0.05$; **, $p < 0.005$; ***, $p < 0.0005$; error bar of SEM.

Additionally, we believe with proper adaptation, dynamic models could be used in human brain research to explain the single-trial variability. Several researches have reported the trial-variability observed in human subjects. Shruti[29] found that the visual response to central and lateralized faces showed critical trial-variability modulated by task difficulty and age in infants using electroencephalography (EEG); a fMRI study[30] suggested that spontaneous fluctuations were suppressed by attention in human visual cortex; ongoing EEG phase is also proposed as a trial-by-trial predictor for perceptual and attentional variability[31]. Furthermore, theoretical works have been done in building dynamical theory of consciousness and creativity[32].

However, compared with calcium imaging in mice, application on human brain research exhibits certain difference. First, the spatial granule observed in human brain research is generally coarser. Therefore, the basic nodes considered in the dynamic network is no longer a single neuron, but a population of neurons, and the

dynamical system equation should be adjusted according to the physical reality. Second, human brain imaging could easily cover the whole 3D volume, therefore spatial organization of functional brain areas and identification of spatial-temporal principal components should be of more importance[32]. Moreover, human subjects could be engaged in much more complex cognitive states compared with rodents, even in spontaneous activity. This may add extra challenge in building dynamic models for human subjects.

Q7. The lack of introduction to statistical analysis methods makes it uncertain whether the results are reliable. Please give more details on statistical analysis, including the 2-way ANOVA in Line 311, the Wilcoxon rank sum test in Line 351. The Pairwise Wilkson test in Line 1069.

Response: We thank the reviewer for the suggestion. We have added corresponding section in the Method part of our manuscript to describe the statistical analysis, including the detailed description of ANOVA test and Wilcoxon test.

We have carefully inspected the significance tests we used in our manuscript and decided to change the 2-way ANOVA to one-way ANOVA for contrast factor, and a paired Wilcoxon test for temporal factor, due to the dependency between temporal samplings. The p value remained low after the change. Specifically, we first calculated the populational activity in a 1-sec pre-stimulus time window and a 1-sec post-stimulus time window as a coding populational vector. These populational factors are then used to calculate the pre- and post-stimulation trial deviation from the averaged populational vector, using either Pearson correlation or normalized Euclidean norm distance. The difference of post-and pre-trial variability is considered as a sampling point in the ANOVA test. We consider stimulation type (amplitude in DBS and contrast in visual task) as the grouping factor. The ANOVA test is conducted by *anova1* function using MATLAB for p value. To consider the temporal change of trial variability after the onset of stimulation, we use each pre- and post-stimulation trial variability as a sampling point, and grouped them into 2 groups based on temporal window. A paired one-sided Wilcoxon test is used to evaluate if the difference of the pairs has a non-zero median. The test is conducted using *signrank* function in MATLAB.

Q8. In Figure 5e, the authors used * to indicate $p < 0.5$. Generally, we used 0.05 as the threshold. In addition, I wonder where the authors considered corrections of multiple comparisons.

Response: We thank the reviewer for pointing out the problem. We feel sorry for the mistake in the text. We have corrected the threshold and added Bonferroni correction to our multi-comparison significance test. We divided the threshold by the times of comparison $N=66$. So, the actual threshold we used is *, 7.6×10^{-4} ; **, 7.6×10^{-5} ; ***, 7.6×10^{-6} . The significance remained high after the correction. We have corrected the mistake in the main text of Figure 5 caption and updated Figure 5 with the new threshold.

Q9. Please give more details on the clustering in Line 652, including the samples, features, number of iterations, clustering distance.

Response: We thank the reviewer for the constructive suggestion. In simulation, we treat the second principal component of the simulated neural network as the behavior output. For each trial, the 20 timesteps trace is considered as a 20-dimensional feature. We used a k-means classifier to divide the initial states into 2 groups. The number of clusters is set as $k=2$, Euclidean distance is used as the distance metric, and the maximum iteration is set as 100. We have added the details into the revised manuscript.

“For each trial, the 20 timesteps trace is considered as a 20-dimensional feature. We used a k-means classifier to divide the initial states into 2 groups. The number of clusters is set as $k=2$, Euclidean distance is used as the distance metric, and the maximum iteration is set as 100.”

Q10. Please check the definition of Euclidean norm of the matrix in Line 659.

Response: We thank the reviewer for pointing out the problem. We feel sorry for the mistake in the text, we have corrected the expression of Euclidean norm, $\|R\|_F = \sqrt{\sum_{i,j} R_{ij}^2}$ in the revised manuscript. And we have thoroughly checked our manuscript to avoid similar mistakes.

Q11. Please specify the version of “matlab” in Line 565 and “MATLAB” in Line 703. Also, please keep consistent in the manuscript.

Response: We have revised “MATLAB” in the text to keep consistent in the manuscript.

Q12. Please clearly define and describe the variable Hext in Line 606.

Response: We have added the explanation of the variable H^{ext} in the manuscript.

Q13. In Line 512, the meaning of “as describes18” is not clear. Please correct. In addition, please check the “1 ×1.2cm2” in Line 513, the “30°relative” “20°towards” in Line 529.

Response: We thank the reviewer for the comment. We have deleted “as describes18” in the text. The FOV of RUSH system is 1 ×1.2 cm². The space was added to “30° relative” “20° towards”. We also check the other expression to revise the mistakes.

Reviewer #3

This is a very interesting paper which offers an engaging exploration of mesoscale neuronal granular trial variability in vivo, and complemented by simulations using a nonlinear recurrent network in silico. The authors have conducted extensive experiments on mouse brains and provided robust modeling validation. The manuscript is well-organized, and the results are effectively presented. The study utilizes in vivo imaging through the previously developed RUSH system, which records a large number of neurons at micrometer resolution. The authors have established a comprehensive pipeline to manage the substantial neuronal data collected across more than 12 cortical regions. They designed both electrical and visual stimuli to investigate trial variability at the single-neuron level across the entire dorsal cortex. This study is presented with clarity and efficiency. Overall, this paper achieves a significant milestone that could greatly influence future studies on trial variability. To strengthen the paper further, I have a few minor suggestions:

Response: We sincerely appreciate the reviewer for the positive evaluation of our work and constructive comments on our manuscript. Detailed responses to each comment are listed below.

Q1. Details regarding the post-surgical recovery of the mice are needed. How long after the craniotomy and the implantation of electrical electrodes do the experiments begin? Additionally, it would be helpful to know the lifespan of the mice following electrode implantation and whether the surgeries impact the mice's health or behavior.

Response: We thank the reviewer for the suggestion. Following craniotomy surgery, mice are given a recovery period of at least 7 days. Subsequently, we assess the integrity of the craniotomy window and the quality of calcium signals. Only mice that have fully recovered and are healthy are included in the experiments after this recovery period. To monitor the condition of mice post-electrode implantation, we conduct long-term behavioral observations. Mice have been observed to live for over a year post-surgery without showing any behavioral differences when compared to healthy, non-operated mice. We have added corresponding details in the revised manuscript.

Q2. The claustrum (Cla) nucleus was chosen as the stimulation site. Many researches have studied this nucleus using some in vitro methods. What are the specific reasons for the author to target this nucleus?

Response: We thank the reviewer for the insightful comment. Because our imaging system facilitates cortex-wide imaging. The claustrum is highly connected to numerous cortical regions throughout the brain, receiving inputs from and sending outputs to almost all cortical areas. This positions it uniquely as a potential coordinator or integrator of widespread cortical activities. Given its extensive connectivity, the claustrum presents a fascinating target for exploring how the modulation of one brain area can impact many others, potentially shedding light on the mechanisms underlying global brain functions such as consciousness and attention.

Q3. The authors tried the electrical stimulus and visual stimulus paradigm to test mesoscale trial variability. What is the difference of electrical stimulus and visual stimulus in trial variability studies? Are there any different characters due to the different kinds of input?

Response: We thank the reviewer for the insightful question. The two types of stimuli discussed are fundamentally different (Figure S4, Figure R6). We initially chose electrical stimulation to study trial variability without the effects of subjective consciousness factors such as attention, arousal, and eye movements. This approach minimizes the influence of external random variations during electrical stimulation. Unlike deep brain stimulation (DBS), which does not induce specific responses in targeted areas, visual stimulation is comparatively milder and elicits clearly localized activation in the primary cortex. This contrast highlights the distinct effects and applications of each stimulation method.

Figure R6 | Comparison of DBS stimulus and visual stimulus. (a), Figure a, The trial-averaged neural post-stimulation responses, showing the empirical probability mass function (PMF) and the cumulative distribution function (CDF). **(b),** Brain state trajectory in the first 2 MDS axis, comparing the DBS with the visual stimulation.

Q4. The author has combined mesoscale fluorescence imaging and deep brain stimulus as the protocol for investigating trial variability. However, trial variability could be explained and depicted in a local circuit and using repeated sensory input, using merely conventional electrophysiology or 2-photon microscopy. What is the advantage of using mesoscale imaging, and why the authors chose DBS as the stimulus method in this study?

Response: We thank the reviewer for the insightful comment. Mesoscale imaging offers a significantly larger field of view (FOV) that captures extensive brain structures with single-neuron resolution. This capability facilitates the visualization and examination of interactions or neuronal variations across different brain regions within a single trial. Such an approach is invaluable for studying trial variability through quantitative analysis of brain regions and neuronal activities over time. In this context, DBS was selected as a targeted stimulus in a specific nucleus to mitigate external influences such as attention, arousal, and eye blinking.

Q5. In Fig. 5, a very simple KNN model is used as the decoder to decode the grating orientation. Why is KNN chosen as the decoder when there are better coding strategies?

Response: We thank the reviewer for the comment. We chose KNN as our primary decoding strategy for grating orientation due to several reasons. First, KNN is non-parametric with minimal hyperparameters, providing a straightforward reflection of single-trial deviations. This simplicity allows us to minimize the influence of model capacity when analyzing trial variability. Secondly, KNN imposes minimal assumptions about the data distribution. Unlike linear models that assume data linearity, KNN identifies nearest neighbors and is adept at capturing nonlinear relationships and local structural details. Thirdly, KNN is particularly effective with small training datasets. Unlike methods that require extensive data to perform well, KNN can directly identify the nearest neighbors in small datasets and apply this knowledge to testing datasets without needing extensive training. Overall, KNN provides a direct insight into trial variability relative to the features of interest and is thus aptly suited for investigating trial variability. Employing KNN, we identified several strategies to mitigate trial variability, including focusing on the targeted field of view (FOV), projecting neural space into lower dimensions, and isolating key neurons. We believe these strategies reflect potential neural mechanisms to counteract trial variability.

Q6. Trial variability has long captivated scientists, particularly in the life sciences, where averaging has traditionally been employed to manage it. However, advancements in observational technology promise to significantly transform our analytical approaches. Given these developments, how can we more effectively leverage trial variability in future research?

Response: We thank the reviewer for the great question. Traditionally, trial variability has often been averaged out because the underlying patterns and correlations were not immediately evident. Looking ahead, future research should adopt high-throughput technologies capable of managing large datasets with significant variability, such as large-scale imaging. These technologies offer a wider scope for data analysis and interpretation. Additionally, employing advanced data analysis techniques could effectively utilize the information inherent in trial variability. For instance, integrating more sophisticated statistical methods, such as

mixed-effects models or machine learning algorithms, could allow for the consideration and analysis of variability instead of merely averaging it. Moreover, using real-time analytics to monitor and adjust experiments based on the variability observed in initial trials could further enhance research outcomes.

Q7. In the manuscript, there are inconsistencies in the spacing between “Fig.” and the number. Please ensure uniformity.

Response: We thank the reviewer for pointing out the problem. Now we have checked the spacing between “Fig.” and the number, and keep them uniformity.

Reviewer #4

In the manuscript titled “Mesoscale neuronal granular trial variability in vivo illustrated by nonlinear recurrent network in silico”, the authors investigated trial variability in large-scale neural recordings by combining mesoscale fluorescence imaging and deep brain stimulation (DBS) in mice. They compared the empirical findings with a simulated neural network model to understand the origin and dynamics of trial variability. The study revealed that trial variability is an intrinsic property of complex, nonlinear neural networks and is influenced by factors such as input amplitude, projection range, and the initial state of the animal. The authors demonstrated that DBS shapes neural dynamics, leading to the formation of a stimulus-dependent attractor state, and that imperfectly tuned neurons can form a spatially heterogeneous coding community, with reliable coding achieved through population dynamics. To conduct this research, the authors developed an experimental setup that combines their previously developed mesoscale fluorescence imaging (RUSH) and DBS, enabling simultaneous stimulation and recording of large-scale neural activity at single-neuron resolution. This synergistic system allows for a comprehensive study of neural dynamics under various stimulation conditions. The authors also visualized the energy landscape of the neural system under different stimulation conditions using the simulated network, offering a new perspective on neural dynamics. This approach helps to understand the formation and dissolution of attractor states in response to external perturbations. Furthermore, the authors proposed strategies for improving decoding performance in the presence of trial variability, such as focusing on targeted regions, dimensionality reduction, and identifying reliable coding units. These strategies offer valuable insights for future research in systems neuroscience and the development of brain-computer interfaces. The manuscript is of high quality and is suitable for publication in Nature Communications.

I only have minor comments for the authors to improve the manuscript.

Response: We sincerely appreciate the reviewer for the positive evaluation of our work and careful review on our manuscript. Detailed responses to each comment are listed below.

Q1. The authors could consider discussing the advantages and limitations of DBS compared to other stimulation methods, such as optogenetics or transcranial magnetic stimulation, in the context of studying trial variability and neural dynamics. This comparison would highlight the specific benefits and drawbacks of using DBS for investigating trial variability and neural dynamics.

Response: We thank the reviewer for the suggestion. We have added the advantages and limitations of DBS compared to optogenetics and transcranial magnetic stimulation in the manuscript accordingly.

"Deep Brain Stimulation (DBS) was selected for our study because it offers precise stimulation in a specific brain region, unlike Transcranial Magnetic Stimulation (TMS). The parameters of DBS can be easily adjusted

to meet the specific needs of patients, providing a flexibility not as readily available in optogenetics, which is limited in clinical applications due to genetic modification requirements. Furthermore, DBS's effects are reversible, ceasing once stimulation is discontinued. However, DBS is more invasive than TMS, requiring the implantation of electrodes. Unlike optogenetics, DBS does not allow for targeting specific types of neurons."

Q2. While the authors focus on the claustrum and its projections to the cortex, it would be informative to discuss whether the findings on trial variability and the effectiveness of the proposed decoding strategies are expected to generalize to other brain regions and neural circuits.

Response: We thank the reviewer for the inspiring question. We believe it is promising to generalize our finding of trial variability to other nucleus projecting to the cortex. We have adopted a simple recurrent network model with external input to depict the quench of trial variability after external stimulation, which means that minor assumption on the specific topology or connectome is made in our model. By mildly adjust the projection range and amplitude parameters, our model could potentially be adapted to other nuclei.

Regarding functional decoding strategies, we believe that certain sensory cortexes, such as auditory cortex and barrel cortex may exhibit similar properties as we found in visual cortex. These sensory areas are spatially well-organized on the mouse cortex, with the majority of relevant neurons concentrated in specific functional areas and only a few dispersed outside these regions. In such task-specific areas, our spatial model of task relevant neurons should remain applicable.

However, it is important to note that some cognitive variables, such as choice, action, and engagement, are reported to be globally encoded across the cortex [28, 33]. In scenarios involving distributed encoding, identifying a task-specific coding axis may be more crucial than focusing solely on a specific field of view (FOV) or task-related neurons.

In summary, we believe that the emergence of trial variability in complex networks represents a universal phenomenon in the rodent brain. Nonetheless, specific nuclei or functional cortices may exhibit slight variations in projection range and the spatial distribution of stimulus-projected neurons. We have also analyzed a Neuropixel dataset recorded by Kenneth's lab [27] in the revised version with consistent conclusions. In this dataset, the mice were trained to perform a visual discrimination task, and approximately 30,000 neurons are recorded in 10 mice, 39 sessions.

Q3. What is the computational efficiency of the proposed decoding strategies, particularly when dealing with large-scale neural recordings. The authors can suggest potential optimizations for real-time applications.

Response: We thank the reviewer for the question. We selected neurons based on single-neuron trial variability or utilizing populational features extracted through supervised or unsupervised down-sampling strategies to decode specific sensory variables. The computational cost will depend on the chosen classifier, the number of neurons involved, the number of trials, and the duration of each trial.

In our manuscript, to demonstrate the principal of neuron and feature selection, we have selected a simple KNN as our decoder. In the case of KNN decoder, the response trace of all considered neuron is treated as features of

neural representation. The feature vector of each trial contains $N_{neuron} \times N_{time}$ dimensions in total. And the

main computational cost of KNN comes from calculating the pairwise distance of all trials, influenced by the feature number of each trial. Therefore, by selecting k neurons out of N total neurons (or down-sample the neuron space into a k dimensional feature space), we reduce the feature dimension to $\frac{k}{N}$ of original dimension. This could reduce the computational cost of decoding algorithm in both KNN and other decoders, especially when $N \gg k$. In visual decoding task, we found the order of magnitude of $\frac{k}{N}$ between 10^{-2} to 10^{-3} , thus leading to a significant computational cost saving.

For real-time application, we suggest a 2-step decoding procedure. Initially, simple sensory stimulation tests can be used to identify neurons that consistently respond to a specific variable with minimal trial-to-trial variability, or to establish a relevant stimulus-related feature dimension. Subsequently, utilizing these pre-identified neurons or predefined neural dimensions can significantly enhance the signal-to-noise ratio (SNR) of neural readouts. A pre-trained decoder can then be employed for real-time, high-fidelity decoding.

Q4: Grammar:

a. Line 294, change "trial variability follows an approximate exponential decrease function" to "trial variability follows an approximately exponential decrease function".

Response: We thank the reviewer for pointing out the problem. We have revised the expression in text.

b. Line 220, change "The distribution of the z-scored neural fluorescence intensity changes is centered at zeros" to "The distribution of the z-scored neural fluorescence intensity changes is centered at zero".

Response: We have revised the expression accordingly in main text.

c. Add missing articles (a/an/the) throughout the manuscript.

Response: We have checked and revised the expression in text and adding missing articles (a/an/the).

d. Use either "in silico" or "in-silico" consistently throughout the manuscript.

Response: We have checked and revised "in silico".

Reference:

1. Wu, J., et al., *Iterative tomography with digital adaptive optics permits hour-long intravital observation of 3D subcellular dynamics at millisecond scale*. Cell, 2021. **184**(12): p. 3318-3332 e17.
2. Sompolinsky, H., A. Crisanti, and H.J. Sommers, *Chaos in random neural networks*. Phys Rev Lett, 1988. **61**(3): p. 259-262.
3. Rajan, K., L.F. Abbott, and H. Sompolinsky, *Stimulus-dependent suppression of chaos in recurrent neural networks*. Physical Review E, 2010. **82**(1).
4. Andalman, A.S., et al., *Neuronal Dynamics Regulating Brain and Behavioral State Transitions*. Cell, 2019. **177**(4): p. 970-985 e20.
5. Mante, V., et al., *Context-dependent computation by recurrent dynamics in prefrontal cortex*. Nature, 2013. **503**(7474): p. 78-84.

6. Rajan, K., C.D. Harvey, and D.W. Tank, *Recurrent Network Models of Sequence Generation and Memory*. Neuron, 2016. **90**(1): p. 128-42.
7. Nair, A., et al., *An approximate line attractor in the hypothalamus encodes an aggressive state*. Cell, 2023. **186**(1): p. 178-193 e15.
8. Rizzoli, S.O. and W.J. Betz, *Synaptic vesicle pools*. Nat Rev Neurosci, 2005. **6**(1): p. 57-69.
9. Deborah E. Cabin, K.S., Kellie L. McIlwain, Bonnie Orrison, et al., *Synaptic Vesicle Depletion Correlates with Attenuated Synaptic Responses to Prolonged Repetitive Stimulation in Mice Lacking α -Synuclein*. The Journal of Neuroscience, 2002. **22**(20): p. 11.
10. Stevens†, C.R.a.C.F., *Definition of the Readily Releasable Pool of Vesicles at Hippocampal Synapses*. Neuron, 1996. **16**: p. 10.
11. Zucker, R.S., *Short-Term Synaptic Plasticity*. Annual Review of Neuroscience, 1989. **12**(Volume 12, 1989): p. 13-31.
12. Wesseling, C.F.S.a.J.F., *Activity-Dependent Modulation of the Rate at which Synaptic Vesicles Become Available to Undergo Exocytosis*. Neuron, 1998. **21**: p. 9.
13. CHARLES F. STEVENS, A.T.T., *Estimates for the pool size of releasable quanta at a single central synapse and for the time required to refill the pool*. Proc. Natl. Acad. Sci. USA. **92**: p. 4.
14. Schoonover, C.E., et al., *Representational drift in primary olfactory cortex*. Nature, 2021. **594**(7864): p. 541-546.
15. Marks, T.D. and M.J. Goard, *Stimulus-dependent representational drift in primary visual cortex*. Nat Commun, 2021. **12**(1): p. 5169.
16. *Deep brain stimulation during sleep enhances human brain synchrony and memory*. Nat Neurosci, 2023. **26**(6): p. 930-931.
17. Nobauer, T., et al., *Video rate volumetric Ca(2+) imaging across cortex using seeded iterative demixing (SID) microscopy*. Nat Methods, 2017. **14**(8): p. 811-818.
18. Rummyantsev, O.I., et al., *Fundamental bounds on the fidelity of sensory cortical coding*. Nature, 2020. **580**(7801): p. 100-105.
19. Stringer, C., et al., *High-precision coding in visual cortex*. Cell, 2021. **184**(10): p. 2767-2778 e15.
20. Yoshida, T. and K. Ohki, *Natural images are reliably represented by sparse and variable populations of neurons in visual cortex*. Nat Commun, 2020. **11**(1): p. 872.
21. Miller, J.E., et al., *Visual stimuli recruit intrinsically generated cortical ensembles*. Proc Natl Acad Sci U S A, 2014. **111**(38): p. E4053-61.
22. Jesús Pérez- Ortega*, T.A.-G., Rafael Yuste, *Long- term stability of cortical ensembles*. elife, 2021. **10**:e64449: p. 1-19.
23. van den Boom, B.J.G., et al., *Unraveling the mechanisms of deep-brain stimulation of the internal capsule in a mouse model*. Nature Communications, 2023. **14**(1).
24. Xiao, G., et al., *Dopamine and Striatal Neuron Firing Respond to Frequency-Dependent DBS Detected by Microelectrode Arrays in the Rat Model of Parkinson's Disease*. Biosensors-Basel, 2020. **10**(10).
25. Xiao, G., et al., *Microelectrode Arrays Modified with Nanocomposites for Monitoring Dopamine and Spike Firings under Deep Brain Stimulation in Rat Models of Parkinson's*

- Disease*. *Acs Sensors*, 2019. **4**(8): p. 1992-2000.
26. Zhang, Y., et al., *High frequency stimulation of subthalamic nucleus synchronously modulates primary motor cortex and caudate putamen based on dopamine concentration and electrophysiology activities using microelectrode arrays in Parkinson's disease rats*. *Sensors and Actuators B-Chemical*, 2019. **301**.
 27. Rajamani, N., et al., *Deep brain stimulation of symptom-specific networks in Parkinson's disease*. *Nat Commun*, 2024. **15**(1): p. 4662.
 28. Steinmetz, N.A., et al., *Distributed coding of choice, action and engagement across the mouse brain*. *Nature*, 2019. **576**(7786): p. 266-273.
 29. Naik, S., et al., *Event-related variability is modulated by task and development*. *NeuroImage*, 2023. **276**.
 30. Broday-Dvir, R., et al., *Quenching of spontaneous fluctuations by attention in human visual cortex*. *NeuroImage*, 2018. **171**: p. 84-98.
 31. Vanrullen, R., et al., *Ongoing EEG Phase as a Trial-by-Trial Predictor of Perceptual and Attentional Variability*. *Front Psychol*, 2011. **2**: p. 60.
 32. Rabinovich, M.I., M.A. Zaks, and P. Varona, *Sequential dynamics of complex networks in mind: Consciousness and creativity*. *Physics Reports*, 2020. **883**: p. 1-32.
 33. Allen, W.E., et al., *Global Representations of Goal-Directed Behavior in Distinct Cell Types of Mouse Neocortex*. *Neuron*, 2017. **94**(4): p. 891-907 e6.

Manuscript: NCOMMS-24-24504-B

Title: Mesoscale neuronal granular trial variability *in vivo* illustrated by nonlinear recurrent network *in silico*

Oct 14, 2024

We sincerely appreciate the thoughtful comments from the reviewers, which have significantly enhanced the quality of our manuscript. We have prepared a detailed point-by-point response letter. The original reviewers' comments are presented in black, while our responses are highlighted **in blue** for clear presentation.

Point by point responses to reviewers' comments:

REVIEWERS' COMMENTS

Reviewer #2 (Remarks to the Author):

Comments: The authors have been very responsive to my prior concerns and addressed them accordingly. I also thank the authors for their collegial response letter.

Response: We sincerely appreciate the reviewer for the careful review and professional comments on our manuscript.

Reviewer #3 (Remarks to the Author):

Comments: The authors have fully addressed my concerns and comments. I agree with the publication of this paper now.

Response: We sincerely appreciate the reviewer for the careful review and professional comments on our manuscript.

Reviewer #4 (Remarks to the Author):

Comments: The authors have addressed my comments and concerns. I recommend its publication in NC.

Response: We sincerely appreciate the reviewer for the careful review and professional comments on our manuscript.